# Cerebral chemoarchitecture shares organizational traits with brain structure and function

Benjamin Hänisch[1,2,3], Justine Y Hansen[4], Boris C Bernhardt[4], Simon B Eickhoff[1,2], Juergen Dukart[1,2], Bratislav Misic[4], Sofie Louise Valk[1,2,3]*

[1]Institute of Neuroscience and Medicine, Brain and Behaviour, Research Centre Jülich, Jülich, Germany; [2]Institute of Systems Neuroscience, Medical Faculty, Heinrich Heine University Düsseldorf, Düsseldorf, Germany; [3]Otto Hahn Group Cognitive Neurogenetics, Max Planck Institute for Human Cognitive and Brain Sciences, Leipzig, Germany; [4]McConnell Brain Imaging Centre, Montréal Neurological Institute, McGill University, Montréal, Canada

**Abstract** Chemoarchitecture, the heterogeneous distribution of neurotransmitter transporter and receptor molecules, is a relevant component of structure–function relationships in the human brain. Here, we studied the organization of the receptome, a measure of interareal chemoarchitectural similarity, derived from positron-emission tomography imaging studies of 19 different neurotransmitter transporters and receptors. Nonlinear dimensionality reduction revealed three main spatial gradients of cortical chemoarchitectural similarity – a centro-temporal gradient, an occipito-frontal gradient, and a temporo-occipital gradient. In subcortical nuclei, chemoarchitectural similarity distinguished functional communities and delineated a striato-thalamic axis. Overall, the cortical receptome shared key organizational traits with functional and structural brain anatomy, with node-level correspondence to functional, microstructural, and diffusion MRI-based measures decreasing along a primary-to-transmodal axis. Relative to primary and paralimbic regions, unimodal and heteromodal regions showed higher receptomic diversification, possibly supporting functional flexibility.

*For correspondence:
s.valk@fz-juelich.de

Competing interest: The authors declare that no competing interests exist.

## Editor's evaluation

This work provides a valuable structural and functional characterization of the neurotransmitter's spatial distribution heterogeneity in cortical and subcortical regions. The authors report a systematic description and annotation of a new 'layer' of brain organization that has been relatively poorly integrated with the wider neuroimaging literature to date. In sum, this article has the potential to be of great interest to a wide audience in neurosciences.

## Introduction

Uncovering how the anatomy of the human brain supports its function is a long-standing goal of neuroscientific research (**Suárez et al., 2020**). Histological mapping studies found that brain areas vary substantially in cellular composition and established a link between cytoarchitectural and functional diversity (**Brodmann, 1909**; **von Koskinas and Koskinas, 1925**; **Vogt and Vogt, 1919**). Next to cellular composition, the brain's chemoarchitecture, the distribution of neurotransmitter receptor and transporter molecules (NTRM) across the cortical mantle, is a similarly important mode of brain neurobiology. Neurotransmitter receptors show a heterogeneous distribution throughout the cortex, closely related to both vertical (laminar) and horizontal cyto- and myeloarchitectural composition,

as shown using postmortem autoradiographical receptor labeling (*Eickhoff et al., 2007*; *Zilles and Amunts, 2009*). Receptor distributions recapitulate histology-defined cortical areas, but also organize different cortical areas into neurochemical families and further subdivide homogeneous cytoarchitectural regions (*Zilles and Amunts, 2009*; *Zilles and Palomero-Gallagher, 2001*). Changes in localized brain function are reflected by changes in receptor distributions, as demonstrated in the changes of multiple receptor densities at the border between primary (V1) and secondary (V2) visual cortex (*Eickhoff et al., 2008*; *Zilles et al., 2004*). Crucially, brain areas sharing similar functionalities also display similarities in the density profiles of multiple neurotransmitter receptor types, the so-called receptor 'fingerprint' (*Zilles and Amunts, 2009*; *Zilles et al., 2004*; *Zilles et al., 2002*; *Morosan et al., 2005*). For example, receptor fingerprints delineate sensory from association cortices (*Dehaene et al., 2005*) and provide a common molecular basis of areas involved in language comprehension (*Zilles et al., 2015*), strongly indicating receptor fingerprints as key features supporting functional specialization. Therefore, dissecting the brain's chemoarchitectural landscape could be crucial in understanding structure–function links in the human brain. Comprehensive analysis of receptor fingerprints has mostly been limited to autoradiography experiments in postmortem brain slices. Recently, multisite efforts agglomerated large-scale open-access datasets of whole-brain NTRM density distributions derived from positron-emission tomography studies, enabling the in vivo study of chemoarchitecture (*Hansen et al., 2022*; *Dukart et al., 2021*). Using this resource, Hansen et al. delineated associations between NTRM density profiles and oscillatory neural dynamics, meta-analytical studies of functional activation, as well as disease-associated cortical abnormality maps. Importantly, they showed that brain regions in the same resting-state functional connectivity (FC) networks as well as structurally connected brain regions display increased chemoarchitectural similarities (*Hansen et al., 2022*), replicating structure–function relationships evident from autoradiography studies (*Zilles and Amunts, 2009*).

These findings, along with the implications of receptor fingerprints in functional specialization, warrant the study of whole-brain, in vivo imaging-derived chemoarchitectural anatomy of the brain. An improved understanding of organizational principles of the neurotransmission landscape could prove critical for basic neuroscience, but also benefit clinical medicine. NTRMs are highly relevant in mental health care, as an extensive body of research links alterations in NTRM expression and distribution patterns to psychiatric diseases (*Nautiyal and Hen, 2017*; *Seeman, 2013*; *Quah et al., 2020*; *Lydiard, 2003*). Additionally, most psychotropic drugs manipulate the brain's neurotransmission landscape and are effective and reliable pillars in the treatment of psychiatric diseases (*Cipriani et al., 2018*; *Huhn et al., 2019*; *Soomro et al., 2008*; *Geddes and Miklowitz, 2013*), although their mechanisms of action are often incompletely understood. Complementary, clinical phenotypes are associated with alterations in multiple neurotransmitter systems (*Moncrieff et al., 2022*; *Kaltenboeck and Harmer, 2018*; *Kesby et al., 2018*). Characterizing the spatial organization of chemoarchitectural features could therefore provide novel avenues toward understanding the neurobiology of psychiatric diseases (*Dean and Keshavan, 2017*; *Harrison et al., 2018*; *Luvsannyam et al., 2022*; *Pauls et al., 2014*).

We furthermore aim to study the anatomy of subcortical chemoarchitecture as the question stands if the relationship between receptor fingerprints and functional specialization observed in the cortex could be generalized to subcortical nuclei (*Zilles and Amunts, 2009*; *Zilles et al., 2015*). Since cortical disparities between functional and structural connectivity could be partly explained by subcortical ascending neuromodulatory projections (*Bell and Shine, 2016*; *Shine, 2019*), a clearer understanding of subcortical chemoarchitecture and its relationship to cortical chemoarchitecture could provide a novel perspective on whole-brain structure–function relationships (*Forstmann et al., 2017*).

Here, we leverage the aforementioned resource published by Hansen et al. to generate and characterize the 'receptome,' a neuroanatomical measure that reflects the interregional similarities of brain regions based on their NTRM fingerprints. To study the spatial organization of chemoarchitectural similarity, we employ an unsupervised dimensionality reduction technique to generate principal gradients, which are low-dimensional representations of the organizational axes in the cortical and subcortical receptome. Using these gradients, we identify NTRM distributions that drive regional receptor (dis)similarity. Several follow-up analyses shed light upon the relationship to organizational axes in structural connectivity (SC), as measured using diffusion MRI (*Yeh et al., 2021*), microstructural profile covariance (MPC) (*Paquola et al., 2019*), and resting-state functional connectivity (rsFC) (*Logothetis, 2008*). Finally, we performed meta-analytic decoding of chemoarchitectural gradients to

assess their relations to topic-based functional brain activation (*Yarkoni et al., 2011*) and investigated their relationship to radiological markers of disease (*Thompson et al., 2014*). We performed various analyses to evaluate the robustness of our observations.

## Results

### Organization of the cortical receptome (Figure 1)

To assess cortical chemoarchitecture, we leveraged a large publicly available dataset of PET-derived NTRM densities, containing 19 different NTRM from a total of over 1200 subjects (*Hansen et al., 2022*). After parcellating the receptor maps into 100 parcels according to the Schaefer atlas (*Schaefer et al., 2018*), we calculated a Spearman rank correlation matrix of parcel-level NTRM densities, the receptome. The receptome represents node-level interregional similarities in NTRM fingerprints. Next, we employed nonlinear dimensionality reduction techniques by leveraging diffusion map embedding to delineate the main organizational axes of cortical chemoarchitectural similarity. A schematic introducing the different NTRM and the workflow is outlined in *Figure 1A*. See Table S1 for a detailed overview of the PET NTRM density maps.

Diffusion embedding-derived gradients showed high correspondence to axes derived by linear dimensionality reduction techniques (*Figure 1—figure supplement 1A*). The first 11 components explained significantly more variance compared to gradients decomposed from receptomes generated from randomized NTRM density maps (*Figure 1—figure supplement 1B*). We chose to focus on the first three gradients, which explained 15, 14, and 13% of relative variance, respectively, due to a marked drop in variance explained after these three components (*Figure 1A*). The first receptome gradient (RC G1) described an axis stretching between somato-motor regions and inferior temporal and occipital lobe. The second receptome gradient (RC G2) spanned between a temporo-occipital and a frontal anchor. Finally, the third receptome gradient (RC G3) was differentiated between the occipital cortex and the temporal lobe (*Figure 1B*).

To determine which NTRM distributions drive the main axes of cortical chemoarchitectural similarity, we performed Spearman rank correlations between a parcel's associated gradient value and its NTRM fingerprint, meaning density profiles of all NTRM in that parcel (*Figure 1C*). Note that the gradient value of a parcel is a measure of where on the gradient axis the parcel is located, from which similarity to parcels with similar values, and dissimilarity to parcels with dissimilar values, is inferred. Thus, a receptor with higher density in parcels with negative values and lower density in parcels with positive values will be negatively correlated to the gradient. RC G1 was primarily driven by the anticorrelation between distributions of 5-HTT, 5-HT4, 5-HT2a, and GABAa with the distributions of VAChT, H3, NAT, and $A4B2$. RC G2 separated 5-HTT, DAT, NMDA, D1, and GABA distributions from α4β2, 5-HT1b, CB1, H3, and MU. RC G3 showed significant negative correlations to GABAa distributions and significant positive correlations to D1, 5-HT1a, CB1, MU, 5-HT4, and VAChT.

### Organization of the subcortical receptome (Figure 2)

Following our analysis of cortical NTRM similarity, we investigated the chemoarchitecture of subcortical nuclei. We selected the caudate nucleus, putamen, nucleus accumbens, pallidal globe, thalamus, and amygdala as regions of interest (ROIs). To gain an understanding of how different the cerebral cortex and subcortical nuclei are in their chemoarchitectural composition, we performed a multidimensional scaling projection of cortical and subcortical NTRM density profiles that were z-scored across both compartments (*Figure 2—figure supplement 1A*). Subcortical nuclei were shown to be largely separate from cortical structures, with the exception of amygdala. NTRM density profiles z-scored only within subcortical nuclei were used in subsequent analyses.

First, to investigate whether NTRM fingerprints in subcortical nuclei were associated with functional specialization, as observed in cortical areas, we performed agglomerative hierarchical clustering on the z-scored mean NTRM density profiles of subcortical ROIs per hemisphere (*Figure 2A*). Subcortical chemoarchitecture was largely symmetrical between hemispheres, as indicated by the immediate clustering of structures with their counterpart from the other hemisphere. The main hierarchical branch separated putamen, accumbens nucleus, caudate nucleus (the striatum), and pallidum from amygdala and thalamus. Thalamus and striatum had considerable differences in NTRM co-expression patterns. α4β2, NAT, 5-HTT, and NMDA showed strong co-expression in thalamus but not in striatum,

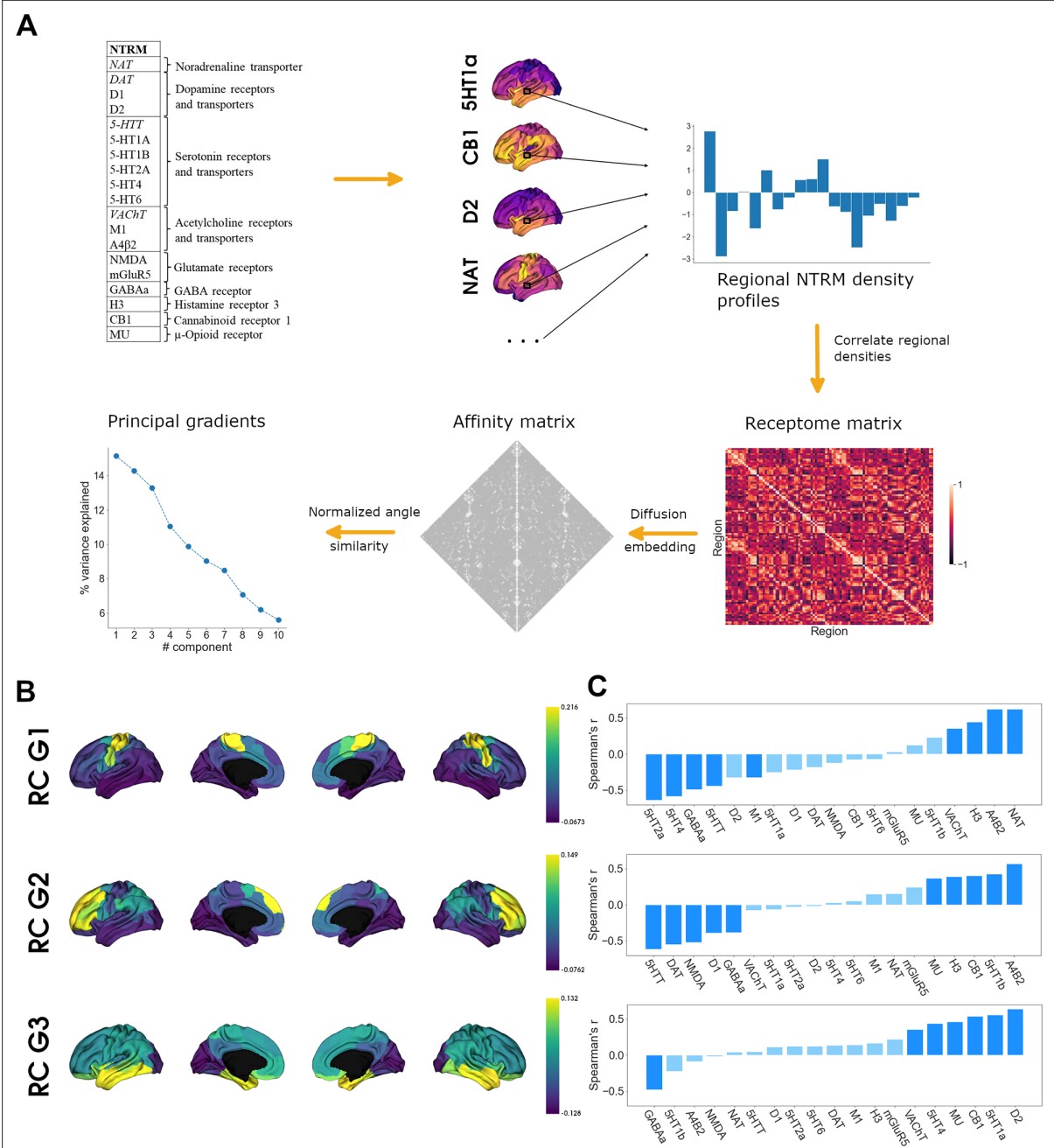

**Figure 1.** Organization of the cortical receptome. (**A**) Analytic workflow of receptome generation and gradient decomposition. Node-level neurotransmitter receptor and transporter molecule (NTRM) fingerprints are derived from PET images of 19 different NTRM (in the top left, italic font denotes transporters). The fingerprints are then Spearman rank correlated to capture node-level similarity in chemoarchitectural composition, generating the receptome matrix. Next, to determine similarity between all rows of the receptome matrix, we used a normalized angle similarity kernel to generate an affinity matrix. Finally, we employ diffusion embedding, a nonlinear dimensionality reduction technique, to derive gradients of receptomic organization. (**B**) Receptome (RC) gradients projected on the cortical surface. Top: first receptome gradient (RC G1); middle: second receptome gradient (RC G2); bottom: third receptome gradient (RC G3). (**C**) Spearman rank correlations of cortical receptome gradients with individual NTRM densities. Top: first receptome gradient; middle: second receptome gradient; bottom: third receptome gradient. Saturated blue coloring corresponds to statistically significant correlations at $p < 0.05$.

The online version of this article includes the following figure supplement(s) for figure 1:

**Figure supplement 1.** Cortical receptome gradients.

**Figure supplement 2.** Robustness of receptome gradients.

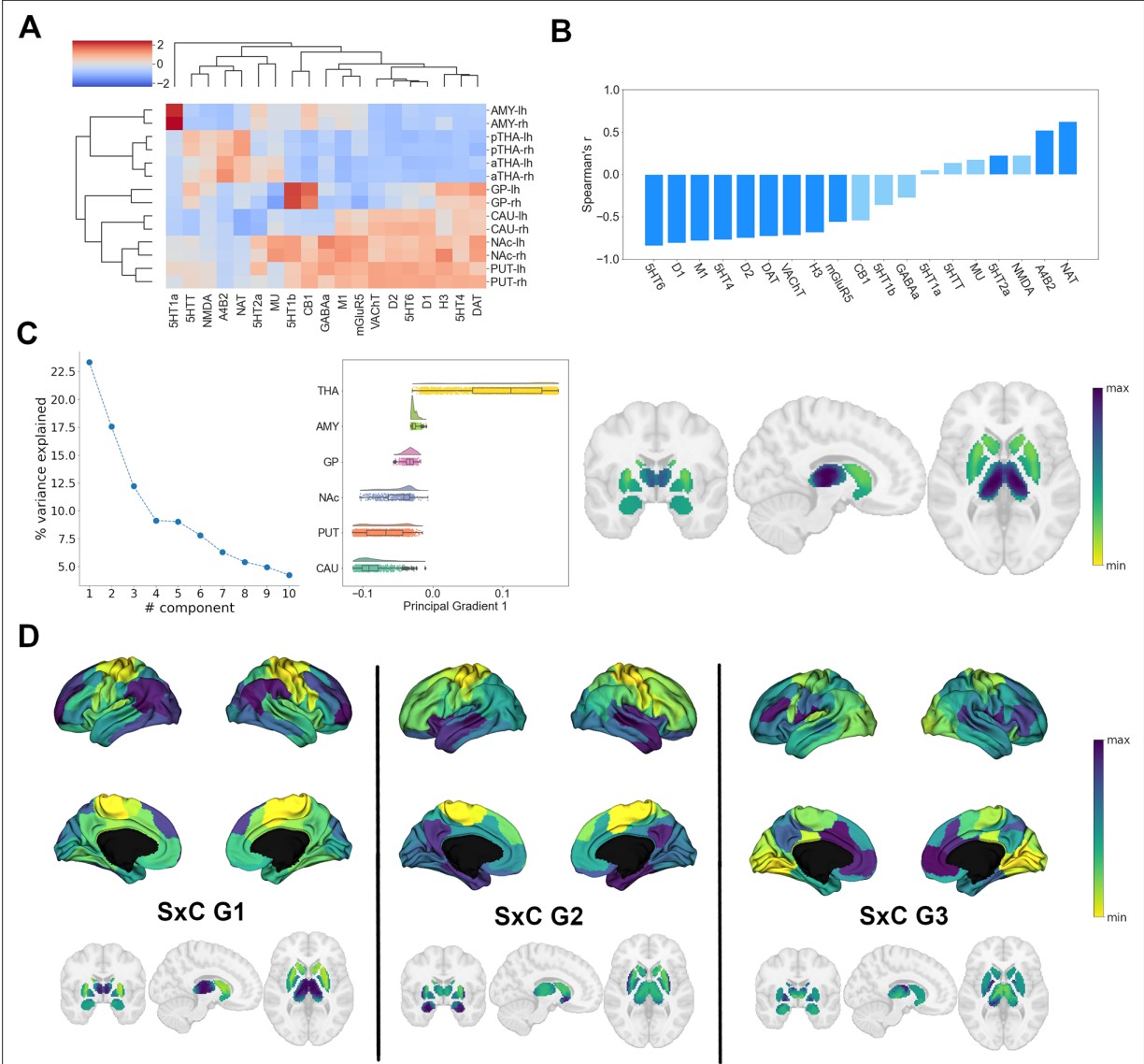

**Figure 2.** Organization of subcortical chemoarchitecture. (**A**) Hierarchical agglomerative clustering of neurotransmitter receptor and transporter molecule (NTRM) densities in subcortical structures. aTHA: anterior thalamus; pTHA: posterior thalamus. (**B**) Spearman rank correlations of the first subcortical receptome gradient with individual NTRM densities. Saturated blue coloring corresponds to statistically significant correlations at $p < 0.05$. (**C**) Gradient decomposition of the subcortical receptome. Left: percentage of variance explained by components following gradient decomposition. Middle: value distribution of the first subcortical receptome gradient across subcortical structures. CAU: caudate nucleus; PUT: putamen; NAc: accumbens nucleus; GP: pallidal globe; AMY: amygdala; THA: thalamus. Right: subcortical projection of the first subcortical receptome gradient. (**D**) Gradients of the subcortico-cortical receptome projected to the cortical surface and to subcortical nuclei.

The online version of this article includes the following figure supplement(s) for figure 2:

**Figure supplement 1.** Subcortical receptome.

**Figure supplement 2.** Robustness of agglomerative hierarchical clustering – subcortex.

while D1, D2, DAT, 5-HT4, 5-HT6, M1, and VAChT were strongly co-expressed in striatum, but not in thalamus.

Then, we analyzed chemoarchitectural similarity in subcortical nuclei through constructing a receptome by voxel-wise Spearman rank correlations of NTRM density profiles in the subcortical ROIs. To discern how subcortical nuclei can be reconstructed based on chemoarchitectural similarity, we employed the Leiden community detection method (**Traag et al., 2019**), a greedy optimization algorithm that opts to minimize variance within and maximize variance between communities.

Subcortical receptome clustering exhibited high stability across the resolution parameter sample space (*Figure 2—figure supplement 1A*). Receptomic clustering discerned three dominant communities, the first mainly capturing the striatal structures (putamen, caudate, NAc) and the pallidal globe, the second mainly capturing the thalamus, and the third mainly capturing the amygdala (*Figure 2—figure supplement 1A*). We then used diffusion embedding to derive low-dimensional gradient embeddings of the subcortical receptome to discern its main organizational axes. The first subcortical receptome gradient (sRC G1), explaining 23% of relative variance, was anchored between the striatum and the thalamus (*Figure 2C*). Note that proximity of structures was not a major determinant of sRC G1 values, demonstrated by voxels of the caudate nucleus and thalamus that were proximal to each other but showed diverging sRC G1 values. The second gradient, explaining 17.5% of relative variance, and third gradient, explaining 12% of relative variance, described ventral-dorsal and medial-lateral trajectories, respectively (*Figure 2—figure supplement 1*). The first subcortical receptome gradient showed significant positive correlations to NAT, α4β2, and 5-HT2a densities, and significant negative correlations to 5-HT6, D1, M1, 5-HT4, D2, DAT, VAChT, H3, and mGluR5 distributions (*Figure 2B*).

Lastly, we were interested in the relationship between the subcortical and cortical receptomes. We created a subcortico-cortical NTRM covariance matrix and applied diffusion embedding to delineate the gradients of subcortico-cortical chemoarchitectural similarity (*Figure 2D*). The first and second cortical gradients correlated significantly with all subcortico-cortical receptome gradients, while the third cortical gradient only correlated significantly to the third subcortico-cortical gradient (*Figure 2—figure supplement 1D*).

## Relationship of the cortical receptome to brain functional processing and disease (Figure 3)

After characterizing the cortical and subcortical receptomes, we sought to investigate the relationship of chemoarchitectural similarity to hallmarks of brain functional processing and dysfunction. To assess brain functional processing, we used topic-based meta-analytical maps of task-based functional brain activation. This approach associates data-driven semantic topics with localized brain activity (e.g. 'primary somatomotor' is associated with activation in the precentral gyrus). Using the Neurosynth database (*Yarkoni et al., 2011*), we calculated Spearman rank correlations between normalized activation maps and receptome gradients while accounting for spatial autocorrelation (*Figure 3B*). Negative correlations imply a relationship between topic-based functional activations mainly located in parcels with negative gradient values. RC G1 showed strong positive correlations with meta-analytical topics of sensory-motor function (topics 2, 17, and 32) and control (topics 16 and 20). Its strongest negative correlations were to topics capturing facial and emotion recognition (topic 40) as well as categorizing and abstract functions (topic 38). RC G2 displayed positive correlations to topics of control (topics 16, 20, and 48) and memory (topic 9), differentiating them from topics of facial and emotion recognition (topic 40) and categorizing and abstract functions (topic 38), with which it showed negative correlations. Lastly, RC G3 showed positive correlations of note to topics related to language and speech (topics 6 and 46) compared to negative correlations to topics of attention and task performance (topics 15 and 47), memory (topic 9), and mental imagery (topic 41).

Secondly, we investigated the association between chemoarchitectural organization and neurodevelopmental conditions or disorders. We leveraged disease-related cortical thickness alterations, a radiological marker of structural abnormalities, derived via a standardized multisite effort (*Thompson et al., 2014*). Cortical thickness was quantified by Cohen's d case-vs.-control effect size and accessed through the ENIGMA toolbox (*Larivière et al., 2021*). We selected autism spectrum disorder (ASD) (*van Rooij et al., 2018*), attention-deficit hyperactivity disorder (ADHD) (*Hoogman et al., 2019*), bipolar disorder (BPD) (*Hibar et al., 2018*), DiGeorge syndrome (22q11.2 deletion syndrome) (DGS) (*Sun et al., 2020*), epilepsy (EPS) (*Whelan et al., 2018*), major depressive disorder (MDD) (*Schmaal et al., 2017*), obsessive compulsive disorder (OCD) (*Boedhoe et al., 2018*), and schizophrenia (SCZ) (*van Erp et al., 2018*) to cover a broad spectrum of diseases (*Figure 3C*).

Receptome gradients captured disease-specific cortical thickness alteration patterns. RC G1 showed positive correlations to the cortical thickness profile of OCD, while RC G2 had negative correlations to cortical thickness alterations in BPD. Both OCD and BPD were primarily associated with cortical thinning, thus, cortical thickness in OCD was reduced where RC G1 values were positive,

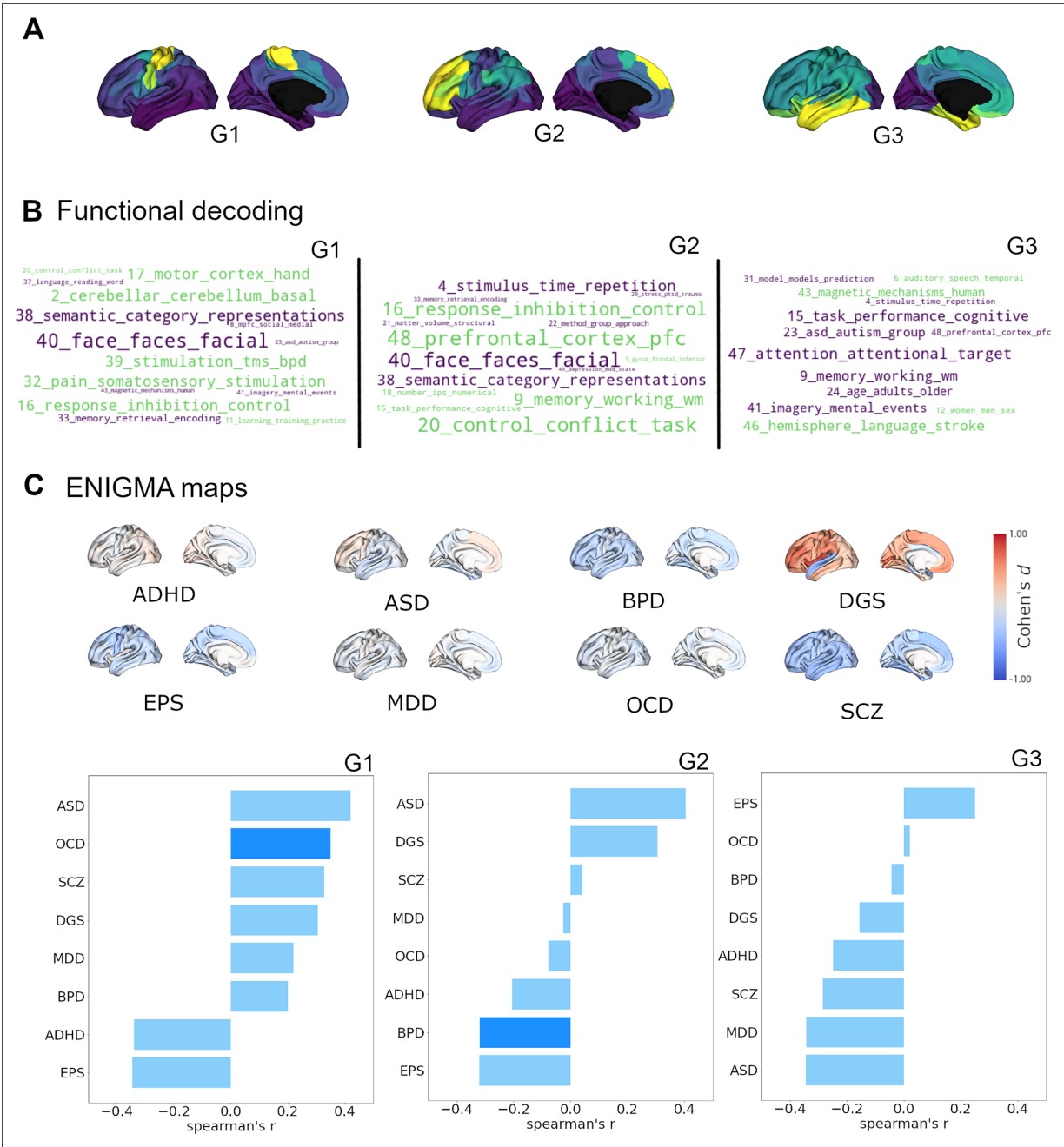

**Figure 3.** Cortical receptome gradients in term-based functional activation and disorder. (**A**) Cortical receptome gradients projected to the cortical surface. (**B**) Functional decoding of cortical receptome gradients. Wordclouds display positive and negative correlations of receptome gradients and topic-based functional activation patterns. Word sizes encode absolute correlation strength, word colors are matched to the respective gradient poles. Only statistically significant correlations (p<0.05) are displayed. Left: RC G1; middle: RC G2; right: RC G3. (**C**) Disease decoding of cortical receptome gradients. Surface plots: effect size (Cohen's d) of cortical thickness alterations in central nervous system disorders in patients vs. controls. Bar plots: Spearman rank correlations of receptome gradients and cortical thickness alterations. Saturated blue coloring corresponds to statistically significant correlations at p < 0.05. Left: RC G1; middle: RC G2; right: RC G3.

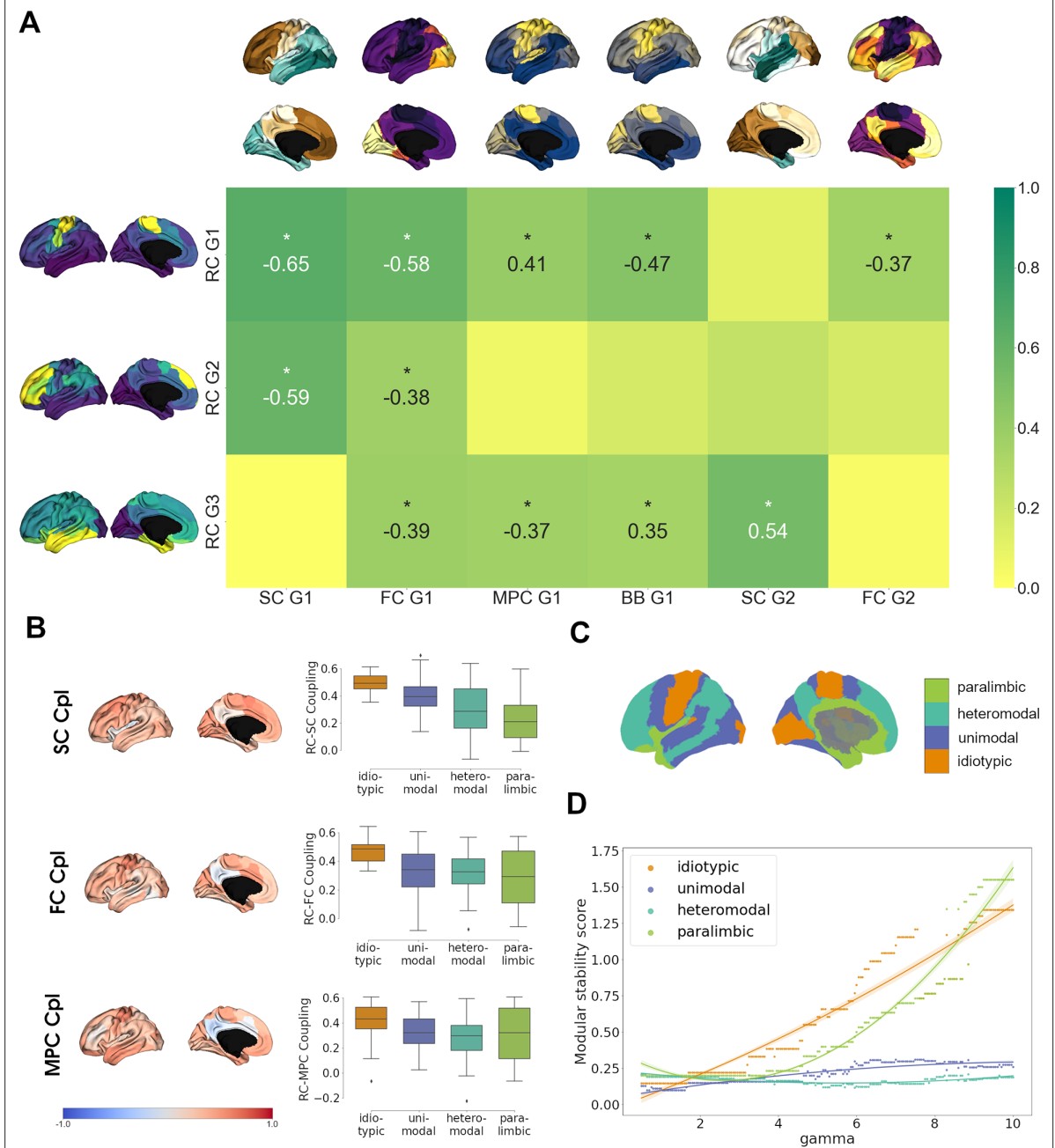

**Figure 4.** Multimodal contextualization of the cortical receptome. (**A**) Correlation strengths of cortical receptome gradients to functional connectivity (FC), structural connectivity (SC), microstructural profile covariance (MPC), and BigBrain gradients. Coloring is scaled to absolute values. Surface-projected gradients are displayed next to their respective rows and columns. Asterisks indicate statistically significant correlations at p < 0.05. (**B**) Coupling of the cortical receptome to SC, FC, and MPC. Left: surface projection of coupling strengths. Right: coupling strengths across cytoarchitectural classes. (**C**) Surface projection of Mesulam cytoarchitectural classes. (**D**) Modular stability of receptome clustering in Mesulam cytoarchitectural classes, reflecting the heterogeneity of receptomic profile.

The online version of this article includes the following figure supplement(s) for figure 4:

**Figure supplement 1.** Contextualization of receptome gradients in hierarchical brain organization.

**Figure supplement 2.** Robustness of agglomerative hierarchical clustering – cortex.

and BPD-associated reductions in cortical thickness were located where RC G2 values were negative. RC G3 did not show significant associations with cortical disease profiles. (*Figure 3C*).

## Interrelationship between the cortical receptome and structural, functional, and cytoarchitectural organization (Figure 4)

Finally, we investigated the relationship of cortical chemoarchitectural similarity to other measures of cortical organization. We first analyzed whether functional brain networks (*Thomas Yeo et al., 2011*) significantly aligned along receptome gradients by comparing gradient value distributions inside functional networks against 1000 random gradient maps generated via variogram matching (*Figure 4—figure supplement 1*). RC G1 showed alignment to the somato-motor network that forms its positive anchor. RC G2 was aligned to default mode and control networks, which are located in the positively anchoring regions, and the visual network, which is located on the opposite side of the gradient. Lastly, RC G3 was aligned with limbic and visual networks, which are located at opposite poles of the gradient.

Then, we aimed to perform a broad multimodal contextualization of cortical chemoarchitectural anatomy. As autoradiography studies connect receptor distributions to cytoarchitectural characteristics (*Zilles and Amunts, 2009*), we compared cortical receptomic organization to MPC, an MRI-derived proxy measure of cortical microstructure (*Foit et al., 2022*), and a gradient of cytoarchitectural variation from the BigBrain project (*Paquola et al., 2019*; *Amunts et al., 2013*) (BB G1). Additionally, we explored the relationships of cortical chemoarchitectural similarity to diffusion MRI tractography-derived SC, and functional MRI-derived resting-state FC, as previous results linked chemoarchitectural similarity to the physical and functional interconnectedness of brain regions (*Hansen et al., 2022*).

We first aimed to compare gradients between these architectural modalities and focused on the first two gradients of SC and FC, and the first gradient of MPC due to the respective amounts of variances explained. RC G1 showed strongest overlaps to SC G1 and FC G1 as these gradients shared either anterior-posterior or visual-to-somatomotor trajectories (*Figure 4A*). Additional weaker correlations were observed with BB G1 and MPC G1, which represent the main axes of cortical cytoarchitectural similarity (*Paquola et al., 2019*), and FC G2, which separates unimodal from association cortices (*Margulies et al., 2016*). Functional network decoding revealed that RC G1 separates visuo-limbic from somatomotor cortices (*Figure 4—figure supplement 1*). Similar to the first receptome gradient, RC G2 correlated significantly to SC G1 and FC G1, while separating visuo-limbic from control networks (*Figure 4—figure supplement 1*). RC G3 showed the strongest correlations to SC G2, which separated occipital from temporal cortex. Further significant correlations existed with FC G1, MPC G1, and BB G1. Functional network decoding placed visual and limbic networks on opposite ends of RC G3 (*Figure 4—figure supplement 1*).

After comparing main anatomical axes, we investigated node-level similarities between the receptome and FC, SC and MPC. We performed row-wise correlations of the receptome matrix to each other matrix (*Figure 4B*). The resulting correlation coefficients expressed the strength of coupling between two measures. Generally, coupling strength of the receptome to the other measures decreased along a sensory-fugal gradient of laminar differentiation, an influential theoretical framework that attributes cognitive processing complexity to cortical areas using cytoarchitectural classes (*Mesulam, 1998*). Average coupling strength across cytoarchitectural classes was significantly different across all metrics. RC-SC decoupling along the sensory-fugal gradient (Kruskal–Wallis' $h$ = 24.43, p<0.001) was driven by significantly stronger coupling in idiotypic relative to heteromodal and paralimbic cortices (post hoc Dunn's test with Bonferroni correction p<0.001). RC-FC coupling strengths in idiotypic cortices were significantly increased relative to unimodal, heteromodal, and paralimbic cortices ($h$ = 16.68, p<0.001; Dunn's test p<0.02). Last, RC-MPC decoupling across cytoarchitectural classes ($h$ = 9.16, p<0.05) was primarily reflected by decreased coupling in heteromodal versus idiotypic regions (Dunn's test p<0.02).

As previous decoding results hinted at a relationship between cortical hierarchy and chemoarchitectural characteristics, we last explored cortical receptomic heterogeneity in the context of cytoarchitectural classes (*Mesulam, 1998*). To this end, we leveraged the Leiden community detection algorithm to discover cortical communities of chemoarchitectural similarity. We observed that new communities primarily formed in the frontal cortex when sampling the resolution parameter space, indicating more unique NTRM fingerprints in the frontal cortex. To capture how stably receptomic

communities recapitulate cytoarchitectural classes when increasing the number of receptomic communities detected, we developed the modular stability score (see 'Materials and methods'). A cytoarchitectural class largely covered by a single receptomic community and not increasingly fracturing with an increase in the overall number of communities has a high modular stability score. Overall, paralimbic cortices exhibited modular stability similar to idiotypic cortices, while heteromodal and unimodal regions were less stable (*Figure 4D*), suggesting that idiotypic and paralimbic cortices contain a more homogeneous receptomic profile, while heteromodal and unimodal cortices have a more diverse chemoarchitectural landscape. We made similar observations studying the relationship of receptomic communities to networks of resting-state functional connectivity (*Thomas Yeo et al., 2011*; *Figure 4—figure supplement 1*).

### Robustness analysis

Owing to the spatial resolution of PET NTRM imaging, we chose to present our main findings in the coarse resolution of 100 Schaefer parcels. To assess validity, we replicated our analyses in Schaefer parcellations 200–400 (*Schaefer et al., 2018*). Selecting a finer granularity than 400 parcels was not reasonable due to the limited resolution of PET images (*Moses, 2011*). Receptome gradients showed good replicability across parcellations (*Figure 1—figure supplement 2*), although an increase in parcellation granularity shifted one extreme in RC G1 and RC G2 toward the temporal poles. Notably, for granularities of 200 and 400 parcels, there is a component ranking switch meaning that the pattern captured by RC G1 in the main results is captured by RC G2 in the replication, and vice versa. As gradients of rsFC, SC, and MPC also change as a function of parcellation granularity, we repeated the correlation analyses across different parcellations. The shift toward the temporal pole in RC G1 and G2 led to a clearer separation between one receptome gradient that strongly correlated to SC G1, and another one that significantly correlated to FC G2 in parcellation granularities 200 and 300 (Tables S2A–D). We additionally replicated agglomerative hierarchical clustering using different linkage methods (*Figure 2—figure supplement 2*, *Figure 4—figure supplement 2*).

## Discussion

In the present work, we investigated the chemoarchitectural anatomy of the human cerebral cortex and subcortex through quantification of interregional chemoarchitectural similarity, leveraging PET imaging-derived neurotransmitter transporter and receptor density maps of 19 different molecules. Furthermore, we aimed to associate chemoarchitecture with imaging-derived markers of brain function and dysfunction, as well as other neuroanatomical modes. In sum, we introduce and thoroughly characterize chemoarchitectural similarity as an additional layer of macro-scale brain organization and present novel structure–function associations in the human brain.

A cornerstone technique of our study was the use of a nonlinear dimensionality reduction technique to derive gradients of the receptome, a matrix of interregional chemoarchitectural similarity. For the cortex, we characterized three receptome gradients, which together explain 42% of relative variance in cortical chemoarchitectural similarity, allowing for an insight into the main anatomical axes that account for nearly half of the cortical receptome's differentiation. The first receptome gradient, RC G1, described an axis stretching between somato-motor regions, where it aligned significantly with the functional somato-motor network, and inferior temporal and occipital lobe. RC G1 combined key features of structural and functional organization, and established similar relationships between cortices as the organization of structural connections, captured by SC G1, which is likely driven by the distance-dependent nature of cortical wiring (*Markov et al., 2013*). It also captured meaningful variations in cytoarchitecture and functional organization, although these correlations were inconsistent across parcellation granularities. Anchoring cortices of RC G1 on the one end were involved in somato-motor and control functions, and facial recognition and abstraction functions on the other end, as revealed by topic-based functional activation decoding. Finally, RC G1 correlated significantly with cortical thickness alterations patterns associated with OCD. Taken together, the first receptome gradient captures the differences in chemoarchitectural composition between the somatomotor regions and the remaining cortex, with the most pronounced divergence outlined against visual and limbic cortices. This chemoarchitectural divide is most apparent in the NTRM distribution patterns of 5-HTT, 5-HT4, 5-HT2a, GABAa and M1 on the one side, which show high density in the temporal

and occipital cortices, and NAT, α4β2, H3 and VAChT on the other site, which have high pericentral and in the frontal densities. RC G1 furthermore connects NTRM density profiles to morphological changes in OCD, where the relationship to serotonin signaling is particularly interesting. Selective serotonin reuptake inhibitors (SSRIs) target 5-HTT and are the preferred pharmacological intervention to treat OCD (*Soomro et al., 2008*; *Lissemore et al., 2018*). Genetically, 5-HT2a and 5-HTT variants have been identified as risk factors for the development of OCD (*Taylor, 2013*), and OCD patients showed aberrant peripheral 5-HTT and 5-HT2a functionality (*Delorme et al., 2005*). In addition, there is emerging evidence that GABA signaling abnormalities are related to the development of OCD (*Pauls et al., 2014*), although conclusive evidence is lacking.

The second receptome gradient, RC G2, spanned between temporo-occipital and frontal anchors, separating the chemoarchitectural composition of visual and limbic networks from attention and control networks. This gradient separated 5-HTT, DAT, NMDA, D1, and GABAa from MU, H3, CB1, 5-HT1b, and α4β2. It correlated significantly to FC G1 and SC G1. Topic-based functional activation decoding revealed that RC G2 spanned between regions linked to abstraction as well as facial and emotion recognition on the one end and regions involved in control and memory on the other end. Moreover, it associated cortical morphological alterations in BPD with features of NTRM fingerprints, where 5-HTT, DAT, and NMDA co-expression is of note. These NTRM have been implicated in genesis and treatment of BPD (*Ghasemi et al., 2014*; *Ashok et al., 2017*; *Pinsonneault et al., 2011*; *Rao et al., 2019*). Lastly, the third receptome gradient, RC G3, was anchored between occipital and temporal cortices. It separated GABAa density distribution patterns from D2, 5-HT1a, CB1, MU, 5-HT4, and VAChT. It correlated significantly to SC G2, FC G1, and gradients of cytoarchitectural differentiation. Functional topic-based decoding revealed that it separated regions involved in auditory and language processing from regions involved in attention, memory, and mental imagery. The separation of visual from limbic cortices distinguished RC G3 from the other two receptome gradients, where limbic and visual cortices were closely aligned.

As both RC G1 and RC G2 outline meaningful relationships between NTRM density profiles and disease morphology, chemoarchitectural similarity could provide novel perspectives in the understanding of the neurobiological basis underlying psychiatric diseases. Investigating NTRM fingerprints rather than focusing on single molecules could shed light on the enigmatic mechanism of actions of psychotropic drugs, especially when taking into account that most take effect through binding multiple types and classes of receptor molecules (*Sullivan et al., 2015*; *Moraczewski and Aedma, 2022*; *Thase, 2008*). However, our results also replicate associations between OCD and BPD and 5-HTT density patterns uncovered using different methodology on the same dataset, further indicating a relevance of this singular molecule in these diseases (*Hansen et al., 2022*). Moreover, both RC G1 and RC G2 capture variations in chemoarchitectural similarity between unimodal and transmodal regions. A separation of sensory from association cortices using their architectural features is possible in multiple modes of architecture (*Paquola et al., 2019*; *Margulies et al., 2016*). The relevance of receptor fingerprints in differentiating sensory from association areas is in line with recent work that employed component analysis to autoradiography-derived receptor densities (*Goulas et al., 2021*). This correspondence across methodological approaches is important as PET imaging is of considerably lower resolution and cannot pick up on cortical layering as an important determinant of NTRM density (*Zilles and Amunts, 2009*). Gradient-based analysis indicated that visual and limbic cortices are relevant anchors in cortical chemoarchitectural similarity axes as they are polar at either one (RC G1 and G2) or both anchors of a gradient (RC G3). Hierarchical clustering of average NTRM densities separated both the visual and limbic network from other functional networks, mirroring clustering results obtained via autoradiography (*Zilles and Palomero-Gallagher, 2017*), and indicating more homogeneous chemoarchitectural compositions in these regions that, importantly, show little overlap between them. Summarizing the interrelationships of receptome gradients and brain structure and function, our results suggest that receptor similarity is organized in a fashion that combines organizational principles of cytoarchitectural, structural, and functional differentiation, although interrelationships to structural and functional connectivity and cytoarchitectural variation present themselves differently across parcellation granularities. Incorporating receptor similarity as a novel layer in studies of structure–function relationships could be crucial to discern a governing set of rules in hierarchical brain architecture (*García-Cabezas et al., 2019*).

Analysis of architectural correspondence on the node level showed significant decoupling of SC and FC from chemoarchitectural similarity, particularly in heteromodal and paralimbic regions, whereas primary areas showed the strongest coupling. This suggests that both structure–function as well as interstructural relationships dissociate in regions conveying more abstract cognitive processes such as attention, cognitive control, and memory (*Spreng et al., 2009*; *Smallwood et al., 2012*; *Smallwood et al., 2021*; *Langner et al., 2018*). Previous work showed that structural and functional connectivity is more closely linked in unimodal cortices and exhibits gradual decoupling toward transmodal cortices, a phenomenon that is hypothesized to be instrumental for human flexible cognition (*Preti and Van De Ville, 2019*; *Liu et al., 2022*; *Valk et al., 2022*). Replicating this observation for chemoarchitectural similarity suggests that diversification of NTRM fingerprints may be equally important to enable flexible cognitive functions (*Suárez et al., 2020*). We corroborate this hypothesis through clustering analysis, where functional networks involved in more abstract cognitive functions and heteromodal cortices show greater receptomic diversity, meaning a wider spread of receptor fingerprints represented in them. This is consistent with associative areas showing high segregation into subareas based on their receptor architecture (*Amunts et al., 2010*). High receptomic diversity might be a disease vulnerability factor as recent work has shown that cortical thickness alterations across different diseases are most pronounced in heteromodal cortices (*Hettwer et al., 2022*). However, it has to be noted that primary regions show a lesser degree of interindividual neuroanatomical variability compared to heteromodal regions, which could be a possible methodological confound influencing our finding of sensory-to-fugal architectural decoupling (*Mueller et al., 2013*). Notably, our results exemplify a chemoarchitectural divide between heteromodal and paralimbic cortices as the latter showed NTRM co-distribution homogeneity similar to idiotypic cortices. A mechanistic explanation might be that, next to memory and emotion (*RajMohan and Mohandas, 2007*), olfactory areas are also located in paralimbic cortices, adding a sensory component to their function (*Courtiol and Wilson, 2017*). Additionally, recent work has indicated a differentiation between heteromodal and paralimbic regions, where the former show decreased heritability and cross-species similarity (*Valk et al., 2022*). Further work may focus on uncovering the developmental mechanisms underlying the differentiation between structure and function of these transmodal zones, also taking into account its diverging chemoarchitecture.

Finally, we could expand a chemoarchitecturally driven structure–function relationship observed in the cortex (*Morosan et al., 2005*; *Dehaene et al., 2005*; *Zilles et al., 2015*; *Zilles and Palomero-Gallagher, 2017*) to subcortical nuclei. Hierarchical agglomerative clustering of NTRM fingerprints revealed a meaningful separation of subcortical structures based on their functionality, exemplified by the differentiation of striatal structures (putamen, accumbens, and caudate nuclei) and pallidal globe from thalamus. Striatum and pallidal globe constitute the basal ganglia, which, together with the thalamus, form the cortico-basal ganglia-thalamic loop. Here, basal ganglia are implicated in motor functions and complex signal integration, while the thalamus orchestrates the communication between large-scale cortical networks (*Bell and Shine, 2016*; *Hwang et al., 2017*; *Lanciego et al., 2012*). This functional divide is not only reflected in NTRM fingerprints, but also in receptomic Leiden clustering and gradient decomposition, where the first subcortical receptomic gradient describes a striato-thalamic axis. We observed partial similarity in NTRM fingerprint composition driving subcortical and cortical chemoarchitectural similarity. While differences in co-distribution patterns of 5-HT4 and M1 from α4β2 and NAT were relevant in both cortex and subcortex, the two areas differ in other relevant NTRM co-distribution patterns. For example, 5-HTT and α4β2 distributions in the cortex are prominently anticorrelated but show similar distributions in subcortial nuclei. Irrespective of individual NTRM co-expressions, a general similarity in subcortical and cortical receptome organization is indicated by overlapping cortical and subcortico-cortical receptome gradients. Considering similarities and differences in NTRM fingerprints could be important when investigating the modulating influence of subcortico-cortical projections on functional brain networks (*Bell and Shine, 2016*; *Janacsek et al., 2022*).

## Limitations

It is of note that the resource we used to comprise the receptome, while extensive, does not exhaustively cover all cerebral neurotransmitter systems. Important molecules such as the α2 noradrenaline receptor, which is an important drug target in the central nervous system (*Smith and Elliott, 2001*;

*Alam et al., 2013*), are missing from our dataset. Our findings must be viewed with the incompleteness of our primary resource in mind. Additionally, we want to point out that in assessing chemoarchitectural anatomy we decided to study ionotropic receptors, metabotropic receptors, and transporters within a shared framework as they exert influence over each other in complex synaptic signaling processes. For example, D1 and D2 signaling influence NMDA signaling through cAMP-mediated posttranslational modification of the receptor, directly acting upon its neuromodulatory potential (*Neve et al., 2004*). Similarly, neuromodulation through presynaptic transporters is conjunct with receptor expression. For example, the neuromodulatory potency of 5-HTT depends on the post-synaptic availability of serotonin receptors, which would mediate the effect an inhibition of these molecules via a drug, such as Fluoxetine. We therefore argue that when studying the co-expression of molecules involved in neurotransmission, incorporating different receptor types and transporters is crucial, even though these molecules convey different functionalities and are not interchangeable. Regarding our primary resource, while PET scans were performed on healthy participants, information on medication and medical history was not available for all participants. Therefore, we cannot control for potential medication or disease effects. Additionally, the comparatively low spatial resolution of PET imaging is exacerbated by the group-average nature of our dataset. This especially limits the ability to investigate subcortical structures. For example, the thalamus consists of more than 60 nuclei with distinct cellular composition and diverging functionality (*Fama and Sullivan, 2015*), important properties we cannot pick up on. Other important subcortical structures, for example, the subthalamic nuclei, cannot be confidently studied due to their size, limiting our whole-brain perspective to larger subcortical nuclei. A more detailed analysis of the subcortical receptome will require methods with higher resolution (*Gaudin et al., 2019*). Furthermore, we want to point out that, although we employ structural and functional measures to contextualize our findings about chemoarchitectural anatomy, our results do not allow claims about the influence of these anatomical axes of brain function, or their interaction with structural brain elements. The correlative nature of our results enables both a richer and multifaceted characterization of chemoarchitectural anatomy as well as the formulation of hypotheses about the role of chemoarchitecture in functional specialization, but no causal inferences about how chemoarchitecture influences brain structure and function can be derived from them. Dissecting how manipulations in the chemoarchitectural landscape influence structure and function goes beyond the descriptive scope of the current work.

In sum, our work outlines the organization of chemoarchitectural similarity across the cortex and subcortical structures, yielding an additional layer of brain organization associated with structural and functional measures of brain organization in both health and disease. Considering this layer in future studies could prove important in answering how flexible cognition is supported by its physical substrates. Meeting this ultimate goal will provide new avenues to understand, treat, and prevent psychiatric diseases and lessen both the personal and societal burden posed by mental illnesses.

## Materials and methods
### Receptor similarity matrix generation
To investigate cortical and subcortical receptor similarity, we made use of an open-access PET MRI dataset described previously (*Hansen et al., 2022*). The associated receptors/transporters, tracers, number of healthy participants, ages, and original publications, for which we refer to full methodological details, are listed in Table S1. In brief, images were acquired in healthy participants using best practice imaging protocols recommended for each radioligand (*Nørgaard et al., 2019*) and averaged across participants before being shared. Images were registered to the MNI152 template (2009c, asymmetric). No medication history of participants was available. The accuracy and validity of receptor density as derived from the PET images have been confirmed using autoradiography data, and the mean age of participants was shown to have negligible influence on tracer density values (*Hansen et al., 2022*). The cortical receptor density maps were parcellated to 100, 200, 300, and 400 regions based on the Schaefer parcellation (*Schaefer et al., 2018*), averaging the intensity values per parcel. Subcortical NTRM densities were extracted using a functional connectivity-derived topographic atlas (*Tian et al., 2020*). For tracers where more than one study was included, a weighted average was generated. This resulted in a parcel × 19 matrix of format (parcel × receptor). The intensity values

were z-score normalized per tracer. We then performed parcel × parcel Spearman rank correlation of receptor densities, yielding the receptome, a matrix of interregional NTRM similarity.

## Gradient decomposition

To assess the driving axes of cortical and subcortical architectural covariance organization, we employed gradient decomposition using the brainspace python package (*Vos de Wael et al., 2020*). Gradients are low-dimensional manifold representations that allow for the characterization of main organizational principles of high-dimensional data (*Margulies et al., 2016*). To calculate gradients of cortical NTRM covariance, rsFC, and MPC, the full matrix was used. SC gradients were separately calculated for intrahemispheric connections in both hemispheres using procrustes analysis to align the gradients to increase comparability and subsequently concatenated. We excluded interhemispheric connections due to their biased underdetection in dMRI fiber tracking, which would result in gradient decomposition primarily detecting asymmetric interhemispheric axes that are unlikely to possess neurobiological relevance, but rather reflect the aforementioned bias (*Royer et al., 2022*). To calculate the gradients, the respective input matrices were thresholded at 90% and, using a normalized angle similarity kernel, transformed into a square non-negative affinity matrix. We then applied diffusion embedding (*Coifman and Lafon, 2006*), a nonlinear dimensionality reduction technique, to extract a low-dimensional embedding of the affinity matrix. Diffusion embedding projects network nodes into a common gradient space, where their distance is a function of connection strengths. This means that nodes closely together in this space display either many suprathreshold or few very strong connections, while nodes distant in gradient space display weak to no connections. In diffusion embedding, a parameter $\alpha$ controls the influence of sampling density on the underlying manifold (where $\alpha = 0$ equals no influence and $\alpha = 1$ equals maximal influence). Similar to previous work (*Margulies et al., 2016*), we set $\alpha$ to 0.5 to retain global relations in the embedded space and provide robustness to noise in the original matrix.

## Structural, functional, and microstructural profile covariance data generation

To contextualize receptor similarity organization, we aimed to compare it to SC, resting-state FC, and MPC. The diversity pertaining to age and sociodemographic variables of the subjects in the PET dataset made the selection of matched reference subjects for FC, SC, and MPC analysis infeasible. Instead, we opted for the construction of group-consensus FC, SC, and MPC matrices collected from the same healthy individuals, obtained, and processed in a reproducible pipeline to ultimately provide comparability of the receptome to SC, FC, and MPC measures of reference nature. We therefore chose the Microstructure Informed Connectomics (MICA-MICs) dataset (*Royer et al., 2022*) to obtain FC, SC, and MPC data. MRI data was acquired at the Brain Imaging Centre of the Montreal Neurological Institute and Hospital using a 3T Siemens Magnetom Prisma-Fit equipped with a 64-channel head coil from 50 healthy young adults with no prior history of neurological or mental illnesses (23 women; 29.54 ± 5.62 y). No medication history was available. For each participant, (1) a T1-weighted (T1w) structural scan, (2) multi-shell diffusion-weighted imaging (DWI), (3) resting-state functional MRI (rs-fMRI), and (4) a second T1-weighted scan, followed by quantitative T1 (qT1) mapping. Image preprocessing was performed via micapipe, an open-access processing pipeline for multimodal MRI data (*Cruces et al., 2022*). Individual functional connectomes were generated by averaging rs-fMRI time series within cortical parcels and cross-correlating all nodal time series. Individual structural connectomes were defined as the weighted count of tractography-derived whole-brain streamlines. To estimate individual microstructural profile covariance, 14 equivolumetric surfaces were generated to sample vertex-wise qT1 intensities across cortical depths and subsequently averaged within parcels. Parcel-level qT1 intensity values were cross-correlated using partial correlations while controlling for the average cortical intensity profile. The resulting values were log-transformed to obtain the individual MPC matrices (*Paquola et al., 2019*).

To generate the group-average matrix of each modality, precomputed and pre-parcellated matrices of 50 individual subjects were used. As no PET data was available for the medial wall, the rows and columns representing it in all SC, FC, and MPC matrices were discarded. For SC and FC matrices additionally, rows and columns containing values for subcortical regions were discarded as well as no analysis of subcortical SC and FC was intended. To generate the group-consensus MPC matrix, parcel

values across the subjects were averaged. To generate the group-consensus FC matrix, the subject matrices underwent Fisher's r-to-z transformation, and subsequently, parcel values across the subjects were averaged. To generate the group-consensus SC matrix, individual matrices were log-transformed and parcel values across subjects were averaged. Afterward, we applied distance-dependent thresholding to account for the over-representation of short-range and under-representation of long-range connections in non-thresholded group-consensus SC matrices (*Betzel et al., 2019*), and the resulting thresholded matrix was used in subsequent analyses.

## Coupling analysis

To investigate the coupling between receptor similarity and FC, SC, and MPC, we performed row-wise Spearman rank correlation analyses of the nonzero elements of the respective matrices.

## Leiden clustering

To evaluate whether NTRM similarity intrinsically structures the cortical surface and subcortical structures, we applied the Leiden clustering algorithm (*Traag et al., 2019*). This clustering analysis enables an assessment of how similarity in chemoarchitecture forms anatomical communities, akin to approaches used to reveal resting-state functional networks (*Thomas Yeo et al., 2011*) or parcellations (*Schaefer et al., 2018*). The Leiden algorithm is a greedy optimization method that aims to maximize the number of within-group edges and minimize the number of between-group edges, with the resulting network modularity being governed by the resolution parameter $\gamma$. To incorporate anticorrelations, we used a negative-asymmetric approach, meaning that we aimed to maximize positive edge weights within communities and negative edge weights between communities. To search the feature space, we chose a $\gamma$ range of 0.5–10 in increments of 0.05 for cortical data, calculating 1000 partition solutions per $\gamma$. For subcortical structures, we chose a $\gamma$ range of 1–10 in increments of 0.5, calculating 250 partitions per $\gamma$. To assess partition stability, we calculated the z-rand score for every partition with every other partition per $\gamma$ value and chose the partition with the highest mean z-rand score, indicating highest similarity to all other partitions for the given $\gamma$ (*Steinley, 2004*; *Pedregosa et al., 2023*). Additionally, we calculated the variance of z-rand scores between partitions per $\gamma$. A high mean z-rand score and a low z-rand score variance indicated a stable partition solution.

## Modular stability

To assess the overlap of cytoarchitectural classes and receptomic clustering, we developed the modular stability score. This metric captures how far a predefined ROI, in our case, a functional network or a cytoarchitectural class, matches a Leiden clustering-derived receptomic community. It is calculated as $Cmax \times \left( \frac{1}{Cin \div Ctot} \right) \times s$, where C*max* is the biggest proportion of the ROI is taken up by one clustering-derived receptomic community, C*in* is the number of different receptomic communities represented inside the ROI, C*tot* is the total number of receptomic communities formed at the given resolution parameter, and *s* is the relative size of the ROI. An ROI that is covered by one receptomic community to a large degree and does not contain a relatively large number of receptomic communities, as measured by the proportion of communities inside the region of interest divided by the total number of communities, will display a high modular stability score. As larger ROIs will have a higher number of communities inside them by chance, we normalize by the relative size of the ROI. We then employ the modular stability score to quantify to what degree predefined ROIs break up into different receptomic communities as the clustering-derived network modularity increases as we sample the resolution parameter space. Note that this experimental score has not been used and verified for validity under other conditions.

## Meta-analytic decoding

To assess the relationship between cortical receptome gradients and localized brain functionality, we leveraged meta-analytical, topic-based maps of functional brain activation, derived from the Neurosynth database (*Tor D., 2011*). Using Nimare, we calculated topic-based activation maps of the Neurosynth v5-50 topic release (https://neurosynth.org/analyses/topics/v5-topics-50/), a set of 50 topics extracted from the abstracts in the full Neurosynth database as of July 2018 using Latent Dirichlet Analysis (*Poldrack et al., 2012*). We parcelled the resulting continuous, non-thresholded

activation maps and performed parcel-wise Spearman rank correlations with the cortical receptome gradients.

## Disorder impact

To assess the relationship between receptome gradients and various neurological and psychiatric diseases, we used publicly available multisite summary statistics of cortical thinning published by the ENIGMA Consortium (*Thompson et al., 2014*). Covariate-adjusted case-vs.-control differences, denoted by across-site random-effects meta-analyses of Cohen's d-values for cortical thickness, were acquired through the ENIGMA toolbox python package (*Larivière et al., 2021*). Multiple linear regression analyses were used to fit age, sex, and site information to cortical thickness measures. Before computing summary statistics, raw data was preprocessed, segmented, and parcellated according to the Desikan-Killiany atlas in FreeSurfer (http://surfer.nmr.mgh.harvard.edu) at each site and according to standard ENIGMA quality control protocols (see http://enigma.ini.usc.edu/protocols/imaging-protocols). To assess a diverse range of cerebral illnesses, we included eight diseases in our analysis: ASD (*van Rooij et al., 2018*), ADHD (*Hoogman et al., 2019*), BPD (*Hibar et al., 2018*), DiGeorge-syndrome (22q11.2 deletion syndrome) (DGS) (*Sun et al., 2020*), EPS (*Whelan et al., 2018*), MDD (*Schmaal et al., 2017*), OCD (*Boedhoe et al., 2018*), and SCZ (*van Erp et al., 2018*). Sample sizes ranged from 1272 (ADHD) to 9572 (SCZ). Summary statistics were derived from adult samples, except for ASD, where all age ranges were used.

## Hierarchical clustering

To discern a similarity hierarchy of subcortical structures and cortical networks based on mean NTRM density, we performed agglomerative hierarchical clustering. Initially, a set of n samples consists of m clusters, where m = n. In an iterative approach, the samples that are most similar are combined into a cluster, where after each iteration, there are m – # iteration clusters (*Nielsen, 2016*). This process is repeated until m = 1. We use Euclidean distance to assess the distance between clusters and use the WPGMA method to select the closest pair of subsets (*Sokal et al., 1958*).

## Null models

Assessment of statistical significance in brain imaging data may be biased when not accounting for spatial autocorrelation of brain imaging signals (*Alexander-Bloch et al., 2018*; *Váša and Mišić, 2022*). To generate permuted brain maps that preserve spatial autocorrelation in parcellated data, we resorted to variogram matching (VGM) (*Burt et al., 2020*). Here, we randomly shuffle the input data and then apply distance-dependent smoothing and rescaling to recover spatial autocorrelation. To assess the significance when comparing surface-projected data, we applied spin permutation (*Alexander-Bloch et al., 2018*) to generate randomly permuted brain maps by random-angle spherical rotation of surface-projected data points, which preserves spatial autocorrelation. Parcel values that got rotated into the medial wall, and values from the medial wall that got rotated to the cortical surface, were discarded (*Markello and Misic, 2021*). In each approach, we generated 1000 permuted brain maps.

## Acknowledgements

We thank Nicola Palomero-Gallagher and Thomas Funck for helpful discussions. We acknowledge and thank those who openly shared their data with the neuroscientific community, which enabled us to perform our study, including the PIs involved in the PET scanning, the MICA-MICS dataset, the Neurosynth tool, as well as all members of the ENIGMA consortium working groups.

## Additional information

### Funding

| Funder | Grant reference number | Author |
| --- | --- | --- |
| Max-Planck-Institut für Kognitions- und Neurowissenschaften | Open Access funding | Sofie Louise Valk |
| Helmholtz International BigBrain Analytics & Laboratory | | Justine Y Hansen<br>Boris C Bernhardt<br>Simon B Eickhoff<br>Sofie Louise Valk |
| Natural Sciences and Engineering Research Council of Canada | | Justine Y Hansen<br>Boris C Bernhardt<br>Bratislav Misic |
| Canadian Institutes of Health Research | | Boris C Bernhardt<br>Bratislav Misic |
| Brain Canada Foundation Future Leaders Fund | | Boris C Bernhardt<br>Bratislav Misic |
| Canada Research Chairs | | Bratislav Misic |
| Michael J. Fox Foundation for Parkinson's Research | | Bratislav Misic |
| Sick Kids Foundation | NI17-039 | Boris C Bernhardt |
| Azrieli Center for Autism Research | ACAR-TACC | Boris C Bernhardt |
| Fonds de Recherche du Québec - Santé | | Boris C Bernhardt |
| Tier-2 Canada Research Chairs program | | Boris C Bernhardt |
| Human Brain Project | | Simon B Eickhoff |
| Helmholtz International Lab grant agreement | InterLabs-0015 | Boris C Bernhardt<br>Simon B Eickhoff<br>Sofie Louise Valk |
| Canada First Research Excellence Fund | CFREF Competition 2 | Boris C Bernhardt<br>Simon B Eickhoff<br>Sofie Louise Valk |
| Horizon 2020 | No. 826421 "TheVirtualBrain-Cloud" | Juergen Dukart |
| Canada First Research Excellence Fund | 2015-2016 | Boris C Bernhardt |
| Max Planck Society | Otto Hahn Award | Sofie Louise Valk |

 The funders had no role in study design, data collection and interpretation, or the decision to submit the work for publication.

### Author contributions

Benjamin Hänisch, Conceptualization, Data curation, Investigation, Visualization, Methodology, Writing - original draft, Writing - review and editing; Justine Y Hansen, Data curation, Investigation, Writing - review and editing; Boris C Bernhardt, Software, Methodology, Writing - review and editing; Simon B Eickhoff, Supervision, Writing - review and editing; Juergen Dukart, Bratislav Misic, Data curation, Writing - review and editing; Sofie Louise Valk, Conceptualization, Supervision, Investigation, Visualization, Methodology, Writing - original draft, Writing - review and editing

### Author ORCIDs

Benjamin Hänisch (iD) http://orcid.org/0000-0001-5463-4218

Justine Y Hansen ![ORCID] http://orcid.org/0000-0003-3142-7480
Boris C Bernhardt ![ORCID] http://orcid.org/0000-0001-9256-6041
Simon B Eickhoff ![ORCID] http://orcid.org/0000-0001-6363-2759
Juergen Dukart ![ORCID] http://orcid.org/0000-0003-0492-5644
Bratislav Misic ![ORCID] http://orcid.org/0000-0003-0307-2862
Sofie Louise Valk ![ORCID] http://orcid.org/0000-0003-2998-6849

### Ethics

Human subjects: The current research complies with all relevant ethical regulations as set by The Independent Research Ethics Committee at the Medical Faculty of the Heinrich-Heine-University of Duesseldorf (study number 2018-317). The current data was based on open access resources, and ethic approvals of the individual datasets are available in the original publications of each data source.

### Decision letter and Author response

Decision letter https://doi.org/10.7554/eLife.83843.sa1
Author response https://doi.org/10.7554/eLife.83843.sa2

---

## Additional files

### Supplementary files

• Supplementary file 1. Table S1. Neurotransmitter receptors and transporters included in analyses. BPND, non-displaceable binding potential; VT, tracer distribution volume; Bmax, density (pmol/ml) converted from binding potential or distributional volume using autoradiography-derived densities; SUVR, standard uptake value ratio. Neurotransmitter receptor maps without citations refer to unpublished data. Table adapted from *Hansen et al., 2022*.

• Supplementary file 2. Table S2A. Replication of multimodal receptome gradient contextualization through correlation using a Schaefer granularity of 100 parcels.

• Supplementary file 3. Table S2B. Replication of multimodal receptome gradient contextualization through correlation using a Schaefer granularity of 200 parcels.

• Supplementary file 4. Table S2C. Replication of multimodal receptome gradient contextualization through correlation using a Schaefer granularity of 300 parcels.

• Supplementary file 5. Table S2D. Replication of multimodal receptome gradient contextualization through correlation using a Schaefer granularity of 400 parcels.

• MDAR checklist

### Data availability

All data and software used in this study is openly accessible. PET data is available here. FC, SC and MPC data is available here. ENIGMA data is available through enigmatoolbox. Meta-analytical functional activation data is available through Neurosynth. The code used to perform the analyses can be found here.

The following previously published datasets were used:

| Author(s) | Year | Dataset title | Dataset URL | Database and Identifier |
|---|---|---|---|---|
| Hansen JY, Shafiei G, Markello RD, Smart K, Cox SML, Nørgaard M | 2022 | Mapping neurotransmitter systems to the structural and functional organization of the human neocortex | https://github.com/netneurolab/hansen_receptors | GitHub, hansen_receptors |

*Continued on next page*

*Continued*

| Author(s) | Year | Dataset title | Dataset URL | Database and Identifier |
|-----------|------|---------------|-------------|-------------------------|
| Royer J, Rodríguez-Cruces R, Tavakol S, Larivière S, Herholz P, Li Q, Vos de Wael R, Paquola C, Benkarim O, Park BY, Lowe AJ, Margulies D, Smallwood J, Bernasconi A, Bernasconi N, Frauscher B, Bernhardt BC | 2021 | MICA-MICs: a dataset for Microstructure-Informed Connectomics | https://n2t.net/ark:/70798/d72xnk2wd397j190qv | Canadian Open Neuroscience Platform, 70798/d72xnk2wd397j190qv |

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
