## [Editor Report]

This work provides a valuable structural and functional characterization of the neurotransmitter's spatial distribution heterogeneity in cortical and subcortical regions. The authors report a systematic description and annotation of a new ‘layer’ of brain organization that has been relatively poorly integrated with the wider neuroimaging literature to date. In sum, this article has the potential to be of great interest to a wide audience in neurosciences.

---

## [Decision Letter]

**Decision letter after peer review:**

Thank you for submitting your article "Neurotransmitter transporter/receptor co-expression shares organizational traits with brain structure and function" for consideration by *eLife*. Your article has been reviewed by 2 peer reviewers, and the evaluation has been overseen by a Reviewing Editor and Michael Frank as the Senior Editor. The following individual involved in the review of your submission has agreed to reveal their identity: Vicente Medel (Reviewer #2).

Essential revisions:

1. Providing some more information before the initial splitting of cortical and subcortical analyses – I see the appeal of treating cortex subcortex separately, but because the analyses are based on z-scoring each receptor within each compartment, jumping straight to this split means that one loses information regarding relative receptor levels in cortical and subcortical structures. Perhaps before presenting the organization within the cortex and subcortex, the authors could do an analysis like MDS projection of cortical and subcortical regions based on receptor values z-scored across both. This would give a sense of whether subcortical structures are as a group very different from cortical structures – which provides important context for the analyses of gradients within each compartment and the relationship between these within-gradient compartments.

2. Better contextualizing the current work relative to the Hansen et al. study mentioned above – I think this issue operates at several levels. First – it would be helpful for readers less familiar with the Hansen study to maybe have a little outline schematic figure at the opening of the Results section (e.g. linked to around line 112) that introduces this resource and shows abbreviations for the different receptors. Second, I think there could be more material in the Intro on what was done in the Hansen preprint and the limitations/open questions left by this work that the current paper seeks to tackle with different methods. Third, I would suggest including supplementary figures that show empirically how the maps, receptor, and functional/disease loadings from the cortical analysis here compare with the PLS (Figure 5 Hansen preprint) and mass-univariate (Figure 6 Hansen preprint) results from Hansen et al.

3. Use of non-linear dimension reduction techniques – Prior work in functional connectivity data (including authors from the current manuscript) has suggested that linear dimension reduction methods like PCA perform favorably compared to the non-linear methods used here (Hong et al., NeuroImage, 2020). Can the authors provide more information/motivation – either based on prior work or comparative analyses in these data – for using non-linear methods of dimension reduction rather than simpler and less parametrically complex non-linear methods?

4. Justifying focus on the first 3 RC gradients – It would be good to say more upfront (e.g. around line 117) about the basis for the focus on 3. Is there a null-based test that could give you the number of significant PCs? E.g. spinning each receptor map – making null PCs and then picking the number of true PCs that predict more variance than their respective null PC?

5. Subcortical analyses – It would be good to more explicitly re-state around line 161 that cortical and subcortical analyses were each conducted on z-scored receptor densities within each compartment. Also, I was not clear on what 2A adds if the data permit a voxel-wise treatment as in 2C. Perhaps start with 2C which then provides justification for the averaging by structure to support simplified representation in a subsequently presented (or sup presented) 2A? Around line 196, it would be good to clarify that this is based on correlation across receptor loadings. Also – could the polarity of gradients be set so that the diagonals of Figure 1C are positive? Finally – if I understand correctly – Figure 2F should have corresponding subcortical maps – it would be good to include these with each cortical one.

6. Neurosynth annotation – the authors state that "Briefly, we selected for a range of terms associated with unimodal (e.g. sensory-motor) to transmodal (e.g. information integration, emotion processing, social cognition) functionality, as well as for terms of neurological and psychiatric diseases (e.g. dementia). The full list can be found in Supplement L1.". This curation of neurosynth maps has the capacity to introduce operator bias. There is redundancy between neurosynth terms and some "noise" terms – but these have already been compressed in a fully data-driven manner by the Neurosynth provided "topic" rather than term maps – so I would suggest using topic terms instead.

7. Yeo parcellation annotation – the authors state "We additionally analyzed the distribution of functional networks along receptome gradients (Figure S2C)(61)." Could spin tests be used to test for any statistically significant alignments – which could then be reported in the main text linking to sup fig?

8. Mentioning potential methodological contributions to loosening of inter-modal coupling in heterogeneity- vs. uni-modal cortices – The interpretation the authors provide for their findings around line 433 is important and should be retained. As stated – there is a general pattern of constraint/coherence vs. diversification for many modalities moving from primary to association. However, this transition also follows increasing anatomical variability across individuals, so it's important to mention this as a methodological factor that could contribute to the results.

9. Different treatment of SC matrices for gradient detection – line 512. Wouldn't this exclude information for all interhemispheric connections (notwithstanding the biased under-detection of these by tractography)? It could be good to provide a bit more information here for the differential treatment of SC vs. other matrices.

10. Modular stability score – around line 589. It would good to have a bit more information about the development of this score. What features is the equation trying to capture (it's not obvious to me from reading it – and this may be shared by some other readers). How well tested is this score for performance under different conditions?

11. The analysis presented is sufficient for the general claims. Still, these claims remain at a very general and abstract level that feels it has no possible null or negative result, which raises a problematic question: what possible result is a negative result for the initial general hypothesis? As the brain has a strong low dimensionality, it can be expected that any possible structural component of the brain (in this case, the neurotransmitter's gradient) is correlated with another random structural component beyond its biologically specific contribution to the system. For the case of gene expression, for example, a random gene likely correlates with low-level structural properties of the brain, but does this speaks about their functional or structural relevance beyond just saying that both are structural elements of the brain?

12. The idea of receptors as conceptually similar to transporters, both spatially categorized in their low-dimensionality as "receptome", can be problematic as it assumes they are interchangeable. I understand the dimensionality reduction strategy of the present work, but still, I believe that biological terminology should be accounted for and clearly distinguished, as they represent important different synaptic processes. This is not really a criticism but more an invitation to use the opportunity to expand on this, given the analytic strategy used in the work. For example, one could think that a re-uptake inhibitor such as atomoxetine (which inhibits presynaptic NET) may increase the level of noradrenaline, thus modulating neuron activity. However, this effect will occur if the postsynaptic neuron is susceptible to changes due to the presence of more extracellular noradrenaline, which is dependent on the receptors. From this perspective, transporters' modulatory power also depends on receptors' availability. This is a strong and clear point for the approach of neurotransmitters similarity index and should be further discussed in the draft.

13. Related to the previous point, the use of "expression" in the title and throughout the text is not the correct terminology that should be used for PET mapping and may lead to confusion from the wide brain mapping literature related to gene expression gradients, a gene co-expression analysis.

14. This work feels too similar to Hansen et al. 2022 (Nat Neur), except for the characterization of more than one gradient of the NTRM similarity, as well as the subcortical results and some small methodological differences. Considering this, I believe these differences should be more clearly stated in the overall work and be used as the milestone results of the work while explicitly stating what results are replication and what has been previously shown by the same group. Moreover, the subcortical results are only characterized but not further explored in their functional, structural, and disease-related relevance as it is done for the cortex. Also, a discussion on why considering principal components explaining less than 20% of the variance in a sample of 19 neurotransmitters spatial distributions is relevant.

*Reviewer #1 (Recommendations for the authors):*

This manuscript describes a secondary analysis of the PET dataset compiled and initially analyzed by Hansen et al. (2022, https://www.biorxiv.org/content/10.1101/2021.10.28.466336v2). The work presented here applies gradient detection algorithms to find the principle axes of neurotransmitter receptor/transporter molecule (NTRM) variation. These axes are then systematically annotated for their alignment with diverse other measures of brain organization. The work is valuable for providing a systematic description and annotation of a new "layer" of brain organization that has been relatively poorly integrated with the wider neuroimaging literature to date.

I think the following issues would benefit from further consideration:

Providing some more information before the initial splitting of cortical and subcortical analyses – I see the appeal of treating cortex subcortex separately, but because the analyses are based on z-scoring each receptor within each compartment, jumping straight to this split means that one loses information regarding relative receptor levels in cortical and subcortical structures. Perhaps before presenting the organization within the cortex and subcortex, the authors could do an analysis like MDS projection of cortical and subcortical regions based on receptor values z-scored across both. This would give a sense of whether subcortical structures are as a group very different from cortical structures – which provides important context for the analyses of gradients within each compartment and the relationship between these within-gradient compartments.

Better contextualizing the current work relative to the Hansen et al. study mentioned above – I think this issue operates at several levels. First – it would be helpful for readers less familiar with the Hansen study to maybe have a little outline schematic figure at the opening of the Results section (e.g. linked to around line 112) that introduces this resource and shows abbreviations for the different receptors. Second, I think there could be more material in the Intro on what was done in the Hansen preprint and the limitations/open questions left by this work that the current paper seeks to tackle with different methods. Third, I would suggest including supplementary figures that show empirically how the maps, receptor, and functional/disease loadings from the cortical analysis here compare with the PLS (Figure 5 Hansen preprint) and mass-univariate (Figure 6 Hansen preprint) results from Hansen et al.

Use of non-linear dimension reduction techniques – Prior work in functional connectivity data (including authors from the current manuscript) has suggested that linear dimension reduction methods like PCA perform favorably compared to the non-linear methods used here (Hong et al., NeuroImage, 2020). Can the authors provide more information/motivation – either based on prior work or comparative analyses in these data – for using non-linear methods of dimension reduction rather than simpler and less parametrically complex non-linear methods?

Justifying focus on the first 3 RC gradients – It would be good to say more upfront (e.g. around line 117) about the basis for the focus on 3. Is there a null-based test that could give you a number of significant PCs? E.g. spinning each receptor map – making null PCs and then picking the number of true PCs that predict more variance than their respective null PC?

Subcortical analyses – It would be good to more explicitly re-state around line 161 that cortical and subcortical analyses were each conducted on z-scored receptor densities within each compartment. Also, I was not clear on what 2A adds if the data permit a voxel-wise treatment as in 2C. Perhaps start with 2C which then provides justification for the averaging by structure to support simplified representation in a subsequently presented (or sup presented) 2A? Around line 196, it would be good to clarify that this is based on correlation across receptor loadings. Also – could the polarity of gradients be set so that the diagonals of Figure 1C are positive? Finally – if I understand correctly – Figure 2F should have corresponding subcortical maps – it would be good to include these with each cortical one.

Neurosynth annotation – the authors state that "Briefly, we selected for a range of terms associated with unimodal (e.g. sensory-motor) to transmodal (e.g. information integration, emotion processing, social cognition) functionality, as well as for terms of neurological and psychiatric diseases (e.g. dementia). The full list can be found in Supplement L1.". This curation of neurosynth maps has the capacity to introduce operator bias. There is redundancy between neurosynth terms and some "noise" terms – but these have already been compressed in a fully data-driven manner by the Neurosynth provided "topic" rather than term maps – so I would suggest using topic terms instead.

Yeo parcellation annotation – the authors state "We additionally analyzed the distribution of functional networks along receptome gradients (Figure S2C)(61)." Could spin tests be used to test for any statistically significant alignments – which could then be reported in the main text linking to sup fig?

Mentioning potential methodological contributions to loosening of inter-modal coupling in heterogeneity- vs. uni-modal cortices – The interpretation the authors provide for their findings around line 433 is important and should be retained. As stated – there is a general pattern of constraint/coherence vs. diversification for many modalities moving from primary to association. However, this transition also follows increasing anatomical variability across individuals, so it's important to mention this as a methodological factor that could contribute to the results.

Different treatment of SC matrices for gradient detection – line 512. Wouldn't this exclude information for all interhemispheric connections (notwithstanding the biased under-detection of these by tractography)? It would be good to provide a bit more information here for the differential treatment of SC vs. other matrices.

Modular stability score – around line 589. It would good to have a bit more information about the development of this score. What features is the equation trying to capture (it's not obvious to me from reading it – and this may be shared by some other readers). How well tested is this score for performance under different conditions?

*Reviewer #2 (Recommendations for the authors):*

This is a very interesting paper that shows evidence for the neurotransmitter's similarity map as a new level of brain organization. I have a few major questions and comments below, which do not challenge the main qualitative findings of the paper but may require additional interpretation from the authors. In addition, I find the stated biological motivation a bit unclear, which I also expand a bit on below.

– The analysis presented is sufficient for the general claims. Still, these claims remain at a very general and abstract level that feels it has no possible null or negative result, which raises a problematic question: what possible result is a negative result for the initial general hypothesis? As the brain has a strong low dimensionality, it can be expected that any possible structural component of the brain (in this case, the neurotransmitter's gradient) is correlated with another random structural component beyond its biologically specific contribution to the system. For the case of gene expression, for example, a random gene likely correlates with low-level structural properties of the brain, but does this speaks about their functional or structural relevance beyond just saying that both are structural elements of the brain?

– The idea of receptors as conceptually similar to transporters, both spatially categorized in their low-dimensionality as "receptome", can be problematic as it assumes they are interchangeable. I understand the dimensionality reduction strategy of the present work, but still, I believe that biological terminology should be accounted for and clearly distinguished, as they represent important different synaptic processes. This is not really a criticism but more an invitation to use the opportunity to expand on this, given the analytic strategy used in the work. For example, one could think that a re-uptake inhibitor such as atomoxetine (which inhibits presynaptic NET) may increase the level of noradrenaline, thus modulating neuron activity. However, this effect will occur if the postsynaptic neuron is susceptible to changes due to the presence of more extracellular noradrenaline, which is dependent on the receptors. From this perspective, transporters' modulatory power also depends on receptors' availability. This is a strong and clear point for the approach of neurotransmitters similarity index and should be further discussed in the draft.

– Related to the previous point, the use of "expression" in the title and throughout the text is not the correct terminology that should be used for PET mapping and may lead to confusion from the wide brain mapping literature related to gene expression gradients, a gene co-expression analysis.

– This work feels too similar to Hansen et al. 2022 (Nat Neur), except for the characterization of more than one gradient of the NTRM similarity, as well as the subcortical results and some small methodological differences. Considering this, I believe these differences should be more clearly stated in the overall work and be used as the milestone results of the work while explicitly stating what results are replication and what has been previously shown by the same group. Moreover, the subcortical results are only characterized but not further explored in their functional, structural, and disease-related relevance as it is done for the cortex. Also, a discussion on why considering principal components explaining less than 20% of the variance in a sample of 19 neurotransmitters spatial distributions is relevant.

---

## [Author Response]

Essential revisions:Reviewer #1 (Recommendations for the authors):1. Providing some more information before the initial splitting of cortical and subcortical analyses – I see the appeal of treating cortex subcortex separately, but because the analyses are based on z-scoring each receptor within each compartment, jumping straight to this split means that one loses information regarding relative receptor levels in cortical and subcortical structures. Perhaps before presenting the organization within the cortex and subcortex, the authors could do an analysis like MDS projection of cortical and subcortical regions based on receptor values z-scored across both. This would give a sense of whether subcortical structures are as a group very different from cortical structures – which provides important context for the analyses of gradients within each compartment and the relationship between these within-gradient compartments.

We thank the Reviewer for this comment, as we agree that an analysis assessing the general likeness of chemoarchitecture between cerebral cortex and subcortical nuclei would prove insightful. We addressed the comment by following the Reviewer’s suggestion of performing an MDS projection on NTRM densities z-scored inter-compartmentally as follows:

Results:

“To gain an understanding of how different the cerebral cortex and subcortical nuclei are in their chemoarchitectural composition, we performed a multidimensional scaling projection of cortical and subcortical NTRM density profiles that were z-scored across both compartments (Figure S2A). Subcortical nuclei were shown to be largely separate from cortical structures, with exception of amygdala.”

2. Better contextualizing the current work relative to the Hansen et al. study mentioned above – I think this issue operates at several levels. First – it would be helpful for readers less familiar with the Hansen study to maybe have a little outline schematic figure at the opening of the Results section (e.g. linked to around line 112) that introduces this resource and shows abbreviations for the different receptors. Second, I think there could be more material in the Intro on what was done in the Hansen preprint and the limitations/open questions left by this work that the current paper seeks to tackle with different methods. Third, I would suggest including supplementary figures that show empirically how the maps, receptor, and functional/disease loadings from the cortical analysis here compare with the PLS (Figure 5 Hansen preprint) and mass-univariate (Figure 6 Hansen preprint) results from Hansen et al.

We thank the Reviewer for this helpful comment. Indeed, our study is related to Hansen et al. published in Nature Neuroscience (2022) in terms of its data and multimodal comparisons. However, as the studies ask and answer different questions, we agree it is very important that the two studies are distinguishable. We agree that it is important to further clarify the resource, explain our motivation and research question further and point out comparable results. We addressed the comment as follows:

We introduce the resource in the Results section:

“To assess cortical chemoarchitecture, we leveraged a large publicly available dataset of PET-derived NTRM densities, containing 19 different NTRM from a total of over 1200 subjects(1). After parcellating the receptor maps into 100 parcels according to the Schaefer atlas(2), we calculated a Spearman rank correlation matrix of parcel-level NTRM densities, the receptome. The receptome represents node-level interregional similarities in NTRM fingerprints. Next, we employed non-linear dimension reduction techniques by leveraging diffusion map embedding to delineate the main organizational axes of cortical chemoarchitectural similarity. A schematic introducing the different NTRM and the workflow is outlined in Figure 1A. See Table S1 for a detailed overview of the PET NTRM density maps.”

Additionally, we extended Figure 1 by a schematic that introduces the different NTRM and outlines the workflow used to generate the receptome matrix and generate its key organizational axes.

Furthermore, we adapted Table S1 to contain the abbreviations as well as non-abbreviated NTRM denominations.

Next, we restructured the introduction to outline our research question and the scope of this paper, which is the neuroanatomical characterization of chemoarchitectural similarity. We furthermore summarize the main results of the study by Hansen et al., Nature Neuroscience (2022), and how their findings in combination with the state of research in neurotransmitter receptor mapping shape our research question and warrant our approach.

Introduction:

“Uncovering how the anatomy of the human brain supports its function is a long-standing goal of neuroscientific research(42). Histological mapping studies found that brain areas vary substantially in cellular composition and established a link between cytoarchitectural and functional diversity(43–45). Next to cellular composition, the brain’s chemoarchitecture, the distribution of neurotransmitter receptor and transporter molecules (NTRM) across the cortical mantle, is a similarly important mode of brain neurobiology. Neurotransmitter receptors show a heterogeneous distribution throughout the cortex, closely related to both vertical (laminar) and horizontal cyto- and myeloarchitectural composition, as shown using post mortem autoradiographical receptor labeling (46,47). Receptor distributions recapitulate histology-defined cortical areas, but also organize different cortical areas into neurochemical families and further subdivide homogeneous cytoarchitectural regions(47,48). Changes in localized brain function are reflected by changes in receptor distributions, as demonstrated in the changes of multiple receptor densities at the border between primary (V1) and secondary (V2) visual cortex(49,50). Crucially, brain areas sharing similar functionalities also display similarities in the density profiles of multiple neurotransmitter receptor types, the so-called receptor “fingerprint”(47,50–52). For example, receptor fingerprints delineate sensory from association cortices(53) and provide a common molecular basis of areas involved in language comprehension(54), strongly indicating receptor as key features supporting functional specialization. Therefore, dissecting the brain’s chemoarchitectural landscape could be crucial in understanding structure-function links in the human brain. Comprehensive analysis of receptor fingerprints has mostly been limited to autoradiography experiments in post-mortem brain slices. Recently, multi-site efforts agglomerated large-scale open-access datasets of whole-brain NTRM density distributions derived from Positron-Emission tomography studies, enabling the in vivo study of chemoarchitecture(55,56). Using this resource, Hansen et al. delineated associations between NTRM density profiles and oscillatory neural dynamics, meta-analytical studies of functional activation as well as disease-associated cortical abnormality maps. Importantly, they showed that brain regions in the same resting-state Functional Connectivity (FC) networks as well as structurally connected brain regions display increased chemoarchitectural similarities(55), replicating structure-function relationships evident from autoradiography studies(47).

These findings, along with the implications of receptor fingerprints in functional specialization, warrant the study of whole-brain, in vivo imaging-derived chemoarchitectural anatomy of the brain. An improved understanding of organizational principles of the neurotransmission landscape could prove critical for basic neuroscience, but also benefit clinical medicine. NTRM are highly relevant in mental health care, as an extensive body of research links alterations in NTRM expression and distribution patterns to psychiatric diseases (57–60). Additionally, most psychotropic drugs manipulate the brain’s neurotransmission landscape and are effective and reliable pillars in the treatment of psychiatric diseases(61–64), although their mechanisms of action are often incompletely understood. Complementary, clinical phenotypes are associated with alterations in multiple neurotransmitter systems(65–67). Characterizing the spatial organization of chemoarchitectural features could therefore provide novel avenues towards understanding the neurobiology of psychiatric diseases(68–71).

We furthermore aim to study the anatomy of subcortical chemoarchitecture, as the question stands if the relationship between receptor fingerprints and functional specialization observed in the cortex could be generalized to subcortical nuclei(47,54). Since cortical disparities between functional and structural connectivity could be partly explained by subcortical ascending neuromodulatory projections (72,73), a clearer understanding of subcortical chemoarchitecture and its relationship to cortical chemoarchitecture could provide a novel perspective on whole-brain structure-function relationships(74).

Here, we leverage the aforementioned resource published by Hansen et al. to generate and characterize the “receptome”, a neuroanatomical measure that reflects the interregional similarities of brain regions based on their NTRM fingerprints. To study the spatial organization of chemoarchitectural similarity, we employ an unsupervised dimensionality reduction technique to generate principal gradients, which are low-dimensional representations of the organizational axes in the cortical and subcortical receptome. Using these gradients, we identify NTRM distributions that drive regional receptor (dis)similarity. Several follow-up analyses shed light upon the relationship to organizational axes in structural connectivity (SC), as measured using diffusion MRI (75), Microstructural Profile Covariance (MPC)(76) and resting-state functional connectivity (rsFC)(77). Finally, we performed meta-analytic decoding of chemoarchitectural gradients to assess their relations to topic-based functional brain activation(78), and investigated their relationship to radiological markers of disease(79). We performed various analyses to evaluate the robustness of our observations.”

Last, the Reviewer asked us to include Figures from Hansen et al., Nat. Neur. (2022) in the current work. We addressed overlapping findings regarding Figure 6 in the Discussion as follows:

Discussion:

“Investigating NTRM fingerprints rather than focusing on single molecules could shed light on the enigmatic mechanism of actions of psychotropic drugs, especially when taking into account that most take effect through binding multiple types and classes of receptor molecules (65–67). However, our results also replicate associations between OCD and bipolar disorder and 5-HTT density patterns uncovered using different methodology on the same dataset, further indicating a relevance of this singular molecule in these diseases(14).”

We furthermore addressed overlapping results regarding chemoarchitectural similarity in supplementary figures (See Hansen et al., Nat. Neur. 2022, Figure 2D).

In their Figure 5., Hansen et al. used a PLS analysis to delineate a joint latent variable between NTRM and Neurosynth-derived functional activation term distributions. This is different from our study, where we do not focus on NTRM-to-activation co-distribution, but rather use term-based functional activation data to gain an insight into the cortical functions that are differentiated by receptome gradients, keeping the two modalities separate. We believe that the two analyses are fundamentally different, as they answer different questions through different methodologies. As we believe that in following the Reviewer’s helpful suggestions, we introduced a clear differentiability of the current work from the aforementioned study by Hansen et al., we made the conscious decision to not incorporate figures from the aforementioned study in our manuscript.

3. Use of non-linear dimension reduction techniques – Prior work in functional connectivity data (including authors from the current manuscript) has suggested that linear dimension reduction methods like PCA perform favorably compared to the non-linear methods used here (Hong et al., NeuroImage, 2020). Can the authors provide more information/motivation – either based on prior work or comparative analyses in these data – for using non-linear methods of dimension reduction rather than simpler and less parametrically complex non-linear methods?

We thank the Reviewer for this comment! The primary motivation of using non-linear as opposed to linear dimensionality reduction techniques is to keep methodological consistency when comparing main axes of the receptome to gradients in functional, structural and cytoarchitectural measures, which is important in the multimodal contextualization we perform in this work in comparison to previous results, where especially microstructural and functional gradients are well characterized (76,80–83). In the receptome, there is no marked difference between PCA and diffusion embedding-based dimensionality reduction. We addressed the comment as follows:

In Figure S1, we display the first three PCA-derived receptome axes. These are highly correspondent to diffusion embedding-derived gradients, as can be both visually seen as well as mathematically assessed, where the respective axes correlate with r >.98.

Results:

“Diffusion embedding-derived gradients showed high correspondence to axes derived by linear dimensionality reduction techniques (Figure S1A).”

4. Justifying focus on the first 3 RC gradients – It would be good to say more upfront (e.g. around line 117) about the basis for the focus on 3. Is there a null-based test that could give you the number of significant PCs? E.g. spinning each receptor map – making null PCs and then picking the number of true PCs that predict more variance than their respective null PC?

We thank the Reviewer for this comment! In line with the literature that investigates gradients in the brain (76,84–86), we decided on the number of components to analyze based on the amount of variance the axes explained, and the marked drop in variance explained after the first three components. We addressed the comment by generating VGM-randomized NTRM density maps, which we used to construct receptomes and delineate gradients. This was performed 1000 times. Subsequently, we compared the amounts of variance explained by the gradients from randomized receptomes to the gradients delineated from the true receptome. As the first 11 true components explained significantly more variance than their randomized counterparts, we could not sensibly use the significance level as an indicator of which axes to analyze further in the paper. We instead chose to keep our original reasoning that we outlined above to focus on the first three gradients. The comment was addressed in the manuscript as follows:

Results:

“The first 11 components explained significantly more variance compared to gradients decomposed from receptomes generated from randomized NTRM density maps (Figure S1B). We chose to focus on the first three gradients, which explained 15%, 14% and 13% of relative variance respectively, due to a marked drop in variance explained after these three components (Figure 1A).”

5. Subcortical analyses – It would be good to more explicitly re-state around line 161 that cortical and subcortical analyses were each conducted on z-scored receptor densities within each compartment. Also, I was not clear on what 2A adds if the data permit a voxel-wise treatment as in 2C. Perhaps start with 2C which then provides justification for the averaging by structure to support simplified representation in a subsequently presented (or sup presented) 2A? Around line 196, it would be good to clarify that this is based on correlation across receptor loadings. Also – could the polarity of gradients be set so that the diagonals of Figure 1C are positive? Finally – if I understand correctly – Figure 2F should have corresponding subcortical maps – it would be good to include these with each cortical one.

We are happy to further clarify! We added a more explicit statement alluding to the compartmentalized z-scoring in the current work:

Results:

“Subcortical nuclei were shown to be largely separate from cortical structures, with the exception of amygdala. NTRM density profiles z-scored only within subcortical nuclei were used in subsequent analyses.”

Furthermore, we want to clarify the motivation behind the analysis in Figure 2A. We tested the hypothesis whether NTRM fingerprints in subcortical nuclei separate functional communities akin to the analyses performed by Zilles and colleagues, where they relied on hierarchical agglomerative clustering to make their claims(47,54). We could replicate this characteristic for subcortical nuclei in this study. As it is our opinion that this is an important result, we made the conscious decision to keep it as 2A in the main figure. Afterwards, we analyzed the subcortical receptome in Figure 2B-D, so to keep it consistent, we also made the conscious decision to keep the replication of Zilles’ cortical findings as 2A.

We adjusted Figure 2C so that the polarity of gradient values is positive.

Finally, we added the corresponding subcortical projections of the cortico-subcortical gradients, as the Reviewer suggested.

6. Neurosynth annotation – the authors state that "Briefly, we selected for a range of terms associated with unimodal (e.g. sensory-motor) to transmodal (e.g. information integration, emotion processing, social cognition) functionality, as well as for terms of neurological and psychiatric diseases (e.g. dementia). The full list can be found in Supplement L1.". This curation of neurosynth maps has the capacity to introduce operator bias. There is redundancy between neurosynth terms and some "noise" terms – but these have already been compressed in a fully data-driven manner by the Neurosynth provided "topic" rather than term maps – so I would suggest using topic terms instead.

We thank the Reviewer for this comment, as the Neurosynth analysis had been a topic of increased discussion in the conceptualization of this manuscript precisely for the reasons the Reviewer points out. We followed the suggestion to repeat the Neurosynth analysis using a topic- rather than term-based approach and updated the section and figure accordingly.

Results:

“After characterizing the cortical and subcortical receptomes, we sought to investigate the relationship of chemoarchitectural similarity to hallmarks of brain functional processing and dysfunction. To assess brain functional processing, we used topic-based meta-analytical maps of task-based functional brain activation. This approach associates data-driven semantic topics with localized brain activity (e.g. ‘primary somatomotor’ is associated with activation in the precentral gyrus). Using the Neurosynth database (78), we calculated Spearman rank correlations between normalized activation maps and receptome gradients while accounting for spatial autocorrelation (Figure 3B). Negative correlations imply a relationship between topic-based functional activations mainly located in parcels with negative gradient values. RC G1 showed strong positive correlations with meta-analytical topics of sensory-motor function (topics 2, 17, 32) and control (topics 16,20). Its strongest negative correlations were to topics capturing facial and emotion recognition (topic 40) as well as categorizing and abstract functions (topic 38). RC G2 displayed positive correlations to topics of control (topics 16, 20, 48) and memory (topic 9), differentiating them from topics of facial and emotion recognition (topic 40) and categorizing and abstract functions (topic 38), with which it showed negative correlations. Lastly, RC G3 showed positive correlations of note to topics related to language and speech (topics 6, 46), compared to negative correlations to topics of attention and task performance (topics 15, 47), memory (topic 9) and mental imagery (topic 41).”

Methods:

“To assess the relationship between cortical receptome gradients and localized brain functionality, we leveraged meta-analytical, topic-based maps of functional brain activation, derived from the Neurosynth database(87). Using Nimare, we calculated topic-based activation maps of the Neurosynth v5-50 topic release (https://neurosynth.org/analyses/topics/v5-topics-50/), a set of 50 topics extracted from the abstracts in the full Neurosynth database as of July 2018 using Latent Dirichlet Analysis(88). We parcellated the resulting continuous, non-thresholded activation maps and performed parcel-wise Spearman rank correlations with the cortical receptome gradients.”

7. Yeo parcellation annotation – the authors state "We additionally analyzed the distribution of functional networks along receptome gradients (Figure S2C)(61)." Could spin tests be used to test for any statistically significant alignments – which could then be reported in the main text linking to sup fig?

We thank the Reviewer for this comment and the interesting suggestion. We have tested significance of alignments of receptome gradients to Yeo networks by comparing value distributions of true vs SPM-randomized gradients inside the respective networks at a significance level of p < 0.05. The comment was addressed as follows:

Results:

“Finally, we investigated the relationship of cortical chemoarchitectural similarity to other measures of cortical organization. We first analyzed whether functional brain networks (89) significantly aligned along receptome gradients by comparing gradient value distributions inside functional networks against 1000 random gradient maps generated via variogram matching (Figure S2C). RC G1 showed alignment to the somato-motor network which forms its positive anchor. RC G2 was aligned to default mode and control networks, which are located in the positively anchoring regions, and the visual network, which is located on the opposite side of the gradient. Last, RC G3 was aligned with limbic and visual networks, which are located at opposite poles of the gradient.”

8. Mentioning potential methodological contributions to loosening of inter-modal coupling in heterogeneity- vs. uni-modal cortices – The interpretation the authors provide for their findings around line 433 is important and should be retained. As stated – there is a general pattern of constraint/coherence vs. diversification for many modalities moving from primary to association. However, this transition also follows increasing anatomical variability across individuals, so it's important to mention this as a methodological factor that could contribute to the results.

We thank the Reviewer for this comment, as it raises an important confound in our results that should be discussed for the reader. We addressed it as follows:

Discussion:

“Analysis of architectural correspondence on the node-level showed significant decoupling of SC and FC from chemoarchitectural similarity, particularly in heteromodal and paralimbic regions, whereas primary areas showed the strongest coupling. This suggests that both structure-function as well as interstructural relationships dissociate in regions conveying more abstract cognitive processes such as attention, cognitive control, and memory(82,90–92). Previous work showed that structural and functional connectivity are more closely linked in unimodal cortices and exhibit gradual decoupling towards transmodal cortices, a phenomenon that is hypothesized to be instrumental for human flexible cognition(93–95). Replicating this observation for chemoarchitectural similarity suggests that diversification of NTRM fingerprints may be equally important to enable flexible cognitive functions(42). We corroborate this hypothesis through clustering analysis, where functional networks involved in higher-order, more abstract cognitive functions and heteromodal cortices show greater receptomic diversity, meaning a wider spread of receptor fingerprints represented in them. This is consistent with associative areas showing high segregation into sub-areas based on their receptor architecture(96). High receptomic diversity might be a disease vulnerability factor, as recent work has shown that cortical thickness alterations across different diseases are most pronounced in heteromodal cortices(97). However, it has to be noted that primary regions show a lesser degree of interindividual neuroanatomical variability compared to heteromodal regions, which could be a possible methodological confound influencing our finding of sensory-to-fugal architectural decoupling(98).”

9. Different treatment of SC matrices for gradient detection – line 512. Wouldn't this exclude information for all interhemispheric connections (notwithstanding the biased under-detection of these by tractography)? It could be good to provide a bit more information here for the differential treatment of SC vs. other matrices.

We thank the Reviewer for this comment, as we agree that the differences in generation of SC gradients might need more explanation. We added a concise statement that explains the motivation in the *Methods* section as follows:

Methods:

“To assess the driving axes of cortical and subcortical architectural covariance organization, we employed principal gradient decomposition(80) using the brainspace python package(99). To calculate principal gradients of cortical NTRM covariance, rsFC and MPC, the full matrix was used. SC gradients were separately calculated for intrahemispheric connections in both hemispheres, using procrustes analysis to align the gradients to increase comparability, and subsequently concatenated. We excluded interhemispheric connections due to their biased underdetection in dMRI fiber tracking, which would result in gradient decomposition primarily detecting asymmetric interhemispheric axes that are unlikely to possess neurobiological relevance, but rather reflect the aforementioned bias(100).”

10. Modular stability score – around line 589. It would good to have a bit more information about the development of this score. What features is the equation trying to capture (it's not obvious to me from reading it – and this may be shared by some other readers). How well tested is this score for performance under different conditions?

We thank the Reviewer for this comment, as we agree that a clear description of what the modular stability score describes is needed for an understanding of the results. We expanded upon the score and also added a disclaimer about its experimental nature as follows:

Methods:

Modular stability

“To assess the overlap of cytoarchitectural classes and receptomic clustering, we developed the modular stability score. This metric captures how far a predefined region of interest, in our case, a functional network or a cytoarchitectural class, matches a Leiden clustering-derived receptomic community. It is calculated as Cmax×(1Cin ÷ Ctot)× s, where Cmax is the biggest proportion of the region of interest is taken up by one clustering-derived receptomic community, Cin is the number of different receptomic communities represented inside the region of interest, Ctot is the total number of receptomic communities formed at the given resolution parameter, and s is the relative size of the region of interest. A region of interest that is covered by one receptomic community to a large degree and does not contain a relatively large number of receptomic communities, as measured by the proportion of communities inside the region of interest divided by the total number of communities, will display a high modular stability score. As larger regions of interest will have a higher number of communities inside them by chance, we normalize by the relative size of the region of interest. We then employ the modular stability score to quantify to what degree predefined regions of interest break up into different receptomic communities as the clustering-derived network modularity increases as we sample the resolution parameter space. Note that this experimental score has not been used and verified for validity under other conditions.”*Reviewer #2 (Recommendations for the authors):*

This is a very interesting paper that shows evidence for the neurotransmitter's similarity map as a new level of brain organization. I have a few major questions and comments below, which do not challenge the main qualitative findings of the paper but may require additional interpretation from the authors. In addition, I find the stated biological motivation a bit unclear, which I also expand a bit on below.

We thank the Reviewer for the positive and appreciative feedback on our work and the insightful comments, which we have addressed below.

– The analysis presented is sufficient for the general claims. Still, these claims remain at a very general and abstract level that feels it has no possible null or negative result, which raises a problematic question: what possible result is a negative result for the initial general hypothesis? As the brain has a strong low dimensionality, it can be expected that any possible structural component of the brain (in this case, the neurotransmitter's gradient) is correlated with another random structural component beyond its biologically specific contribution to the system. For the case of gene expression, for example, a random gene likely correlates with low-level structural properties of the brain, but does this speaks about their functional or structural relevance beyond just saying that both are structural elements of the brain?

We thank the Reviewer for this critical comment, as it points out a global weak point of this kind of study – the impossibility to make inferences pertaining to causal relationships between or functional relevance of architectural measures, which, in the end, makes asking hypothesis-driven research questions infeasible in the sense that a falsification or verification of claims is difficult. We took the Reviewers’ comments as a motivation to revise the whole manuscript and to point out that the current work is to be read as a descriptive neuroanatomical study, which may invite further hypothesis-driven research. Furthermore, we addressed the comment in the manuscript’s discussion to clearly state that the current work is not a confirmatory study.

Discussion:

“Furthermore, we want to point out that, although we employ structural and functional measures to contextualize our findings about chemoarchitectural anatomy, our results do not allow claims about the influence of these anatomical axes of brain function, or their interaction with structural brain elements. The correlative nature of our results enables both a richer and multifaceted characterization of chemoarchitectural anatomy as well as the formulation of hypotheses about the role of chemoarchitecture in functional specialization, but no causal inferences about how chemoarchitecture influences brain structure and function can be derived from them. Dissecting how manipulations in the chemoarchitectural landscape influence structure and function goes beyond the descriptive scope of the current work.”

– The idea of receptors as conceptually similar to transporters, both spatially categorized in their low-dimensionality as "receptome", can be problematic as it assumes they are interchangeable. I understand the dimensionality reduction strategy of the present work, but still, I believe that biological terminology should be accounted for and clearly distinguished, as they represent important different synaptic processes. This is not really a criticism but more an invitation to use the opportunity to expand on this, given the analytic strategy used in the work. For example, one could think that a re-uptake inhibitor such as atomoxetine (which inhibits presynaptic NET) may increase the level of noradrenaline, thus modulating neuron activity. However, this effect will occur if the postsynaptic neuron is susceptible to changes due to the presence of more extracellular noradrenaline, which is dependent on the receptors. From this perspective, transporters' modulatory power also depends on receptors' availability. This is a strong and clear point for the approach of neurotransmitters similarity index and should be further discussed in the draft.

We thank the Reviewer for this comment! Indeed, our egalitarian treatment of transporters, iontropic and metabotropic receptors could be criticized. However, we were aiming to precisely make the point the Reviewer alludes to in this comment, which is to voluntarily include all molecules based on their interdependence which has the strong potential to influence synaptic signaling. We addressed the comment as follows in the manuscript:

Discussion:

“Additionally, we want to point out that in assessing chemoarchitectural anatomy, we decided to ionotropic receptors, metabotropic receptors, and transporters within a shared framework, as they exert influence over each other in complex synaptic signaling processes. For example, D1 and D2 signaling influence NMDA signaling through cAMP-mediated posttranslational modification of the receptor, directly acting upon its neuromodulatory potential(101). Similarly, neuromodulation through presynaptic transporters is conjunct with receptor expression. For example, the neuromodulatory potency of 5-HTT depends on the postsynaptic availability of serotonin receptors, which would mediate the effect an inhibition of these molecules via a drug, such as Fluoxetine. We therefore argue that when studying the co-expression of molecules involved in neurotransmission, incorporating different receptor types and transporters is crucial, even though these molecules convey different functionalities and are not interchangeable.”

– Related to the previous point, the use of "expression" in the title and throughout the text is not the correct terminology that should be used for PET mapping and may lead to confusion from the wide brain mapping literature related to gene expression gradients, a gene co-expression analysis.

We thank the Reviewer for raising this important point. We have adjusted the formulations pertaining to this comment throughout the manuscript. Accordingly, we have also adjusted the title:

Cerebral chemoarchitecture shares organizational traits with brain structure and function

– This work feels too similar to Hansen et al. 2022 (Nat Neur), except for the characterization of more than one gradient of the NTRM similarity, as well as the subcortical results and some small methodological differences. Considering this, I believe these differences should be more clearly stated in the overall work and be used as the milestone results of the work while explicitly stating what results are replication and what has been previously shown by the same group. Moreover, the subcortical results are only characterized but not further explored in their functional, structural, and disease-related relevance as it is done for the cortex. Also, a discussion on why considering principal components explaining less than 20% of the variance in a sample of 19 neurotransmitters spatial distributions is relevant.

We thank the Reviewer for this comment. We have taken to restructure and edit the manuscript so that a clear differentiability from the study by Hansen et al., Nat. Neur (2022) is evident. Essentially, we perform a neuroanatomical characterization of the receptome. Hansen et al. made introductory findings into this topic in their study in Nat. Neur (2022), which we point out as well as reason why we expand on them in the introduction:

Introduction:

“Comprehensive analysis of receptor fingerprints has mostly been limited to autoradiography experiments in post-mortem brain slices. Recently, multi-site efforts agglomerated large-scale open-access datasets of whole-brain NTRM density distributions derived from Positron-Emission tomography studies, enabling the in vivo study of chemoarchitecture(55,56). Using this resource, Hansen et al. delineated associations between NTRM density profiles and oscillatory neural dynamics, meta-analytical studies of functional activation as well as disease-associated cortical abnormality maps. Importantly, they showed that brain regions in the same resting-state Functional Connectivity (FC) networks as well as structurally connected brain regions display increased chemoarchitectural similarities(55), replicating structure-function relationships evident from autoradiography studies(47).

These findings, along with the implications of receptor fingerprints in functional specialization, warrant the study of whole-brain, in vivo imaging-derived chemoarchitectural anatomy of the brain. An improved understanding of organizational principles of the neurotransmission landscape could prove critical for basic neuroscience, but also benefit clinical medicine. NTRM are highly relevant in mental health care, as an extensive body of research links alterations in NTRM expression and distribution patterns to psychiatric diseases (57–60). Additionally, most psychotropic drugs manipulate the brain’s neurotransmission landscape and are effective and reliable pillars in the treatment of psychiatric diseases(61–64), although their mechanisms of action are often incompletely understood. Complementary, clinical phenotypes are associated with alterations in multiple neurotransmitter systems(65–67). Characterizing the spatial organization of chemoarchitectural features could therefore provide novel avenues towards understanding the neurobiology of psychiatric diseases(68–71).”

Moreover, the Reviewer points out that the functional, structural, and disease-related relevance of the subcortical results is not assessed as for the cortical results. This is correct – functional contextualization in our analyses of subcortical chemoarchitecture is not performed in the context of FC data, but only in more classical neuroanatomical knowledge, as we relate it to known functional communities in subcortical nuclei (see Figure 2A and Figure S2B,C) and Discussion:

Discussion:

“Hierarchical agglomerative clustering of NTRM fingerprints revealed a meaningful separation of subcortical structures based on their functionality, exemplified by the differentiation of striatal structures (putamen, accumbens and caudate nuclei) and pallidal globe from thalamus. Striatum and pallidal globe constitute the basal ganglia, which, together with the thalamus, form the cortico-basal ganglia-thalamic loop. Here, basal ganglia are implicated in motor functions and complex signal integration, while the thalamus orchestrates the communication between large-scale cortical networks(31,83,84). This functional divide is not only reflected in NTRM fingerprints, but also in receptomic Leiden clustering and principal gradient decomposition, where the first principal subcortical receptomic gradient describes a striato-thalamic axis.”

We chose to limit our assessment of subcortical nuclei to chemoarchitecture, which we brought in connection to the cortex by constructing subcortico-cortical chemoarchitectural similarity gradients. A meaningful multimodal contextualization as performed for cortical chemoarchitecture in Figure 3 and Figure 4 went beyond our expertise. However, we made a clearer statement regarding the research question we pose towards subcortical chemoarchitecture in the Introduction section. We are convinced that the differing level of depth regarding the study of cortical versus subcortical chemoarchitecture does not take away from the importance of the subcortical results in the current work.

Introduction:

“We furthermore aim to study the anatomy of subcortical chemoarchitecture, as the question stands if the relationship between receptor fingerprints and functional specialization observed in the cortex could be generalized to subcortical nuclei(47,54). Since cortical disparities between functional and structural connectivity could be partly explained by subcortical ascending neuromodulatory projections (72,73), a clearer understanding of subcortical chemoarchitecture and its relationship to cortical chemoarchitecture could provide a novel perspective on whole-brain structure-function relationships and further our knowledge about this so far understudied region of the human brain(74).”

Finally, the Reviewer asks for a discussion on why considering principal components explaining less than 20% of the variance in a sample of 19 neurotransmitters spatial distributions is relevant. The components characterized here explain less than 20% of variance each, but together, explain 42% of cortical receptome relative variance. Next to the already existing section in the introduction that outlines the amounts of variance explained, we added a section in the discussion to point this out clearly as follows:

Discussion:

“A cornerstone technique of our study was the use of a nonlinear dimensionality reduction technique to derive gradients of the receptome, a matrix of interregional chemoarchitectural similarity. For the cortex, we characterized three receptome gradients, which together explain 42% of relative variance in cortical chemoarchitectural similarity, allowing for an insight into the main anatomical axes that account for nearly half of the cortical receptome’s differentiation.”

Furthermore, the sum of variance explained by the components characterized here are not strikingly different than low-dimensional axes of architectural modes characterized in other works (76,80,102–104), which makes us confident in arguing that a characterization of the first three low-dimensional axes yields meaningful insights into cortical chemoarchitecture.

Generally, we want to use this opportunity to point out clearly that the whole manuscript underwent a rewriting process to introduce differentiability to the study by Hansen et al., Nat Neur 2022, with the most pronounced differences being in the Introduction and Discussion sections. We thank the Reviewers for their comments that enabled this process, as we think that the manuscript benefitted tremendously from it.

1. Hansen JY, Shafiei G, Markello RD, Smart K, Cox SML, Nørgaard M, et al. Mapping neurotransmitter systems to the structural and functional organization of the human neocortex [Internet]. bioRxiv; 2022 [cited 2022 Apr 4]. p. 2021.10.28.466336. Available from: https://www.biorxiv.org/content/10.1101/2021.10.28.466336v2

2. Schaefer A, Kong R, Gordon EM, Laumann TO, Zuo XN, Holmes AJ, et al. Local-Global Parcellation of the Human Cerebral Cortex from Intrinsic Functional Connectivity MRI. Cereb Cortex N Y N 1991. 2018 Sep 1;28(9):3095–114.

3. Kaller S, Rullmann M, Patt M, Becker GA, Luthardt J, Girbardt J, et al. Test-retest measurements of dopamine D1-type receptors using simultaneous PET/MRI imaging. Eur J Nucl Med Mol Imaging. 2017 Jun;44(6):1025–32.

4. Sandiego CM, Gallezot JD, Lim K, Ropchan J, Lin S fei, Gao H, et al. Reference region modeling approaches for amphetamine challenge studies with [11C]FLB 457 and PET. J Cereb Blood Flow Metab. 2015 Apr;35(4):623–9.

5. Smith CT, Crawford JL, Dang LC, Seaman KL, San Juan MD, Vijay A, et al. Partial-volume correction increases estimated dopamine D2-like receptor binding potential and reduces adult age differences. J Cereb Blood Flow Metab Off J Int Soc Cereb Blood Flow Metab. 2019 May;39(5):822–33.

6. Slifstein M, van de Giessen E, Van Snellenberg J, Thompson JL, Narendran R, Gil R, et al. Deficits in prefrontal cortical and extrastriatal dopamine release in schizophrenia: a positron emission tomographic functional magnetic resonance imaging study. JAMA Psychiatry. 2015 Apr;72(4):316–24.

7. Sandiego CM, Matuskey D, Lavery M, McGovern E, Huang Y, Nabulsi N, et al. The Effect of Treatment with Guanfacine, an Alpha2 Adrenergic Agonist, on Dopaminergic Tone in Tobacco Smokers: An [11C]FLB457 PET Study. Neuropsychopharmacol Off Publ Am Coll Neuropsychopharmacol. 2018 Apr;43(5):1052–8.

8. Zakiniaeiz Y, Hillmer AT, Matuskey D, Nabulsi N, Ropchan J, Mazure CM, et al. Sex differences in amphetamine-induced dopamine release in the dorsolateral prefrontal cortex of tobacco smokers. Neuropsychopharmacol Off Publ Am Coll Neuropsychopharmacol. 2019 Dec;44(13):2205–11.

9. Dukart J, Holiga Š, Chatham C, Hawkins P, Forsyth A, McMillan R, et al. Cerebral blood flow predicts differential neurotransmitter activity. Sci Rep. 2018 Mar 6;8(1):4074.

10. Belfort-DeAguiar R, Gallezot JD, Hwang JJ, Elshafie A, Yeckel CW, Chan O, et al. Noradrenergic Activity in the Human Brain: A Mechanism Supporting the Defense Against Hypoglycemia. J Clin Endocrinol Metab. 2018 Mar 23;103(6):2244–52.

11. Sanchez-Rangel E, Gallezot JD, Yeckel CW, Lam W, Belfort-DeAguiar R, Chen MK, et al. Norepinephrine transporter availability in brown fat is reduced in obesity: a human PET study with [11C] MRB. Int J Obes 2005. 2020 Apr;44(4):964–7.

12. Li C shan R, Potenza MN, Lee DE, Planeta B, Gallezot JD, Labaree D, et al. Decreased norepinephrine transporter availability in obesity: Positron Emission Tomography imaging with (S,S)-[(11)C]O-methylreboxetine. NeuroImage. 2014 Feb 1;86:306–10.

13. Ding YS, Singhal T, Planeta-Wilson B, Gallezot JD, Nabulsi N, Labaree D, et al. PET imaging of the effects of age and cocaine on the norepinephrine transporter in the human brain using (S,S)-[(11)C]O-methylreboxetine and HRRT. Synap N Y N. 2010 Jan;64(1):30–8.

14. Savli M, Bauer A, Mitterhauser M, Ding YS, Hahn A, Kroll T, et al. Normative database of the serotonergic system in healthy subjects using multi-tracer PET. NeuroImage. 2012 Oct 15;63(1):447–59.

15. Baldassarri SR, Park E, Finnema SJ, Planeta B, Nabulsi N, Najafzadeh S, et al. Inverse changes in raphe and cortical 5-HT1B receptor availability after acute tryptophan depletion in healthy human subjects. Synap N Y N. 2020 Oct;74(10):e22159.

16. Gallezot JD, Nabulsi N, Neumeister A, Planeta-Wilson B, Williams WA, Singhal T, et al. Kinetic modeling of the serotonin 5-HT(1B) receptor radioligand [(11)C]P943 in humans. J Cereb Blood Flow Metab Off J Int Soc Cereb Blood Flow Metab. 2010 Jan;30(1):196–210.

17. Matuskey D, Bhagwagar Z, Planeta B, Pittman B, Gallezot JD, Chen J, et al. Reductions in Brain 5-HT1B Receptor Availability in Primarily Cocaine-Dependent Humans. Biol Psychiatry. 2014 Nov 15;76(10):816–22.

18. Murrough JW, Czermak C, Henry S, Nabulsi N, Gallezot JD, Gueorguieva R, et al. The Effect of Early Trauma Exposure on Serotonin Type 1B Receptor Expression Revealed by Reduced Selective Radioligand Binding. Arch Gen Psychiatry. 2011 Sep;68(9):892–900.

19. Murrough JW, Henry S, Hu J, Gallezot JD, Planeta-Wilson B, Neumaier JF, et al. Reduced ventral striatal/ventral pallidal serotonin1B receptor binding potential in major depressive disorder. Psychopharmacology (Berl). 2011 Feb;213(2–3):547–53.

20. Pittenger C, Adams TG, Gallezot JD, Crowley MJ, Nabulsi N, James Ropchan null, et al. OCD is associated with an altered association between sensorimotor gating and cortical and subcortical 5-HT1b receptor binding. J Affect Disord. 2016 May 15;196:87–96.

21. Saricicek A, Chen J, Planeta B, Ruf B, Subramanyam K, Maloney K, et al. Test-retest reliability of the novel 5-HT1B receptor PET radioligand [11C]P943. Eur J Nucl Med Mol Imaging. 2015 Mar;42(3):468–77.

22. Beliveau V, Ganz M, Feng L, Ozenne B, Højgaard L, Fisher PM, et al. A High-Resolution in vivo Atlas of the Human Brain’s Serotonin System. J Neurosci. 2017 Jan 4;37(1):120–8.

23. Radhakrishnan R, Nabulsi N, Gaiser E, Gallezot JD, Henry S, Planeta B, et al. Age-Related Change in 5-HT6 Receptor Availability in Healthy Male Volunteers Measured with 11C-GSK215083 PET. J Nucl Med. 2018 Sep;59(9):1445–50.

24. Radhakrishnan R, Matuskey D, Nabulsi N, Gaiser E, Gallezot JD, Henry S, et al. in vivo 5-HT6 and 5-HT2A receptor availability in antipsychotic treated schizophrenia patients vs. unmedicated healthy humans measured with [11C]GSK215083 PET. Psychiatry Res Neuroimaging. 2020 Jan 30;295:111007.

25. Baldassarri SR, Hillmer AT, Anderson JM, Jatlow P, Nabulsi N, Labaree D, et al. Use of Electronic Cigarettes Leads to Significant Beta2-Nicotinic Acetylcholine Receptor Occupancy: Evidence From a PET Imaging Study. Nicotine Tob Res Off J Soc Res Nicotine Tob. 2018 Mar 6;20(4):425–33.

26. Hillmer AT, Esterlis I, Gallezot JD, Bois F, Zheng MQ, Nabulsi N, et al. Imaging of cerebral α4β2* nicotinic acetylcholine receptors with (−)-[18F]Flubatine PET: Implementation of bolus plus constant infusion and sensitivity to acetylcholine in human brain. NeuroImage. 2016 Nov 1;141:71–80.

27. Naganawa M, Nabulsi N, Henry S, Matuskey D, Lin SF, Slieker L, et al. First-in-Human Assessment of 11C-LSN3172176, an M1 Muscarinic Acetylcholine Receptor PET Radiotracer. J Nucl Med Off Publ Soc Nucl Med. 2021 Apr;62(4):553–60.

28. Aghourian M, Legault-Denis C, Soucy JP, Rosa-Neto P, Gauthier S, Kostikov A, et al. Quantification of brain cholinergic denervation in Alzheimer’s disease using PET imaging with [18F]-FEOBV. Mol Psychiatry. 2017 Nov;22(11):1531–8.

29. Bedard MA, Aghourian M, Legault-Denis C, Postuma RB, Soucy JP, Gagnon JF, et al. Brain cholinergic alterations in idiopathic REM sleep behaviour disorder: a PET imaging study with 18F-FEOBV. Sleep Med. 2019 Jun;58:35–41.

30. Galovic M, Al-Diwani A, Vivekananda U, Torrealdea F, Erlandsson K, Fryer TD, et al. in vivo NMDA receptor function in people with NMDA receptor antibody encephalitis [Internet]. medRxiv; 2021 [cited 2022 Aug 22]. p. 2021.12.04.21267226. Available from: https://www.medrxiv.org/content/10.1101/2021.12.04.21267226v1

31. Galovic M, Erlandsson K, Fryer TD, Hong YT, Manavaki R, Sari H, et al. Validation of a combined image derived input function and venous sampling approach for the quantification of [18F]GE-179 PET binding in the brain. NeuroImage. 2021 Aug 15;237:118194.

32. McGinnity CJ, Hammers A, Riaño Barros DA, Luthra SK, Jones PA, Trigg W, et al. Initial evaluation of 18F-GE-179, a putative PET Tracer for activated N-methyl D-aspartate receptors. J Nucl Med Off Publ Soc Nucl Med. 2014 Mar;55(3):423–30.

33. Smart K, Cox SML, Scala SG, Tippler M, Jaworska N, Boivin M, et al. Sex differences in [11C]ABP688 binding: a positron emission tomography study of mGlu5 receptors. Eur J Nucl Med Mol Imaging. 2019;46(5):1179–83.

34. DuBois JM, Rousset OG, Rowley J, Porras-Betancourt M, Reader AJ, Labbe A, et al. Characterization of age/sex and the regional distribution of mGluR5 availability in the healthy human brain measured by high-resolution [(11)C]ABP688 PET. Eur J Nucl Med Mol Imaging. 2016 Jan;43(1):152–62.

35. Nørgaard M, Beliveau V, Ganz M, Svarer C, Pinborg LH, Keller SH, et al. A high-resolution in vivo atlas of the human brain’s benzodiazepine binding site of GABAA receptors. NeuroImage. 2021 May 15;232:117878.

36. Gallezot JD, Planeta B, Nabulsi N, Palumbo D, Li X, Liu J, et al. Determination of receptor occupancy in the presence of mass dose: [11C]GSK189254 PET imaging of histamine H3 receptor occupancy by PF-03654746. J Cereb Blood Flow Metab Off J Int Soc Cereb Blood Flow Metab. 2017 Mar;37(3):1095–107.

37. D’Souza DC, Cortes-Briones JA, Ranganathan M, Thurnauer H, Creatura G, Surti T, et al. Rapid Changes in CB1 Receptor Availability in Cannabis Dependent Males after Abstinence from Cannabis. Biol Psychiatry Cogn Neurosci Neuroimaging. 2016 Jan 1;1(1):60–7.

38. Hirvonen J, Zanotti-Fregonara P, Umhau JC, George DT, Rallis-Frutos D, Lyoo CH, et al. Reduced cannabinoid CB1 receptor binding in alcohol dependence measured with positron emission tomography. Mol Psychiatry. 2013 Aug;18(8):916–21.

39. Normandin MD, Zheng MQ, Lin KS, Mason NS, Lin SF, Ropchan J, et al. Imaging the cannabinoid CB1 receptor in humans with [11C]OMAR: assessment of kinetic analysis methods, test-retest reproducibility, and gender differences. J Cereb Blood Flow Metab Off J Int Soc Cereb Blood Flow Metab. 2015 Aug;35(8):1313–22.

40. Ranganathan M, Cortes-Briones J, Radhakrishnan R, Thurnauer H, Planeta B, Skosnik P, et al. Reduced Brain Cannabinoid Receptor Availability in Schizophrenia. Biol Psychiatry. 2016 Jun 15;79(12):997–1005.

41. Kantonen T, Karjalainen T, Isojärvi J, Nuutila P, Tuisku J, Rinne J, et al. Interindividual variability and lateralization of μ-opioid receptors in the human brain. NeuroImage. 2020 Aug 15;217:116922.

42. Suárez LE, Markello RD, Betzel RF, Misic B. Linking Structure and Function in Macroscale Brain Networks. Trends Cogn Sci. 2020 Apr 1;24(4):302–15.

43. Brodmann K. Vergleichende Lokalisationslehre der Grosshirnrinde in ihren Prinzipien dargestellt auf Grund des Zellenbaues. Barth; 1909.

44. Economo C von Koskinas, Georg N,. Die Cytoarchitektonik der Hirnrinde des erwachsenen Menschen. 1925.

45. Vogt C, Vogt O. Allgemeine ergebnisse unserer hirnforschung. Vol. 21. JA Barth; 1919.

46. Eickhoff SB, Rottschy C, Zilles K. Laminar distribution and co-distribution of neurotransmitter receptors in early human visual cortex. Brain Struct Funct. 2007 Dec;212(3–4):255–67.

47. Zilles K, Amunts K. Receptor mapping: architecture of the human cerebral cortex. Curr Opin Neurol. 2009 Aug;22(4):331–9.

48. Zilles K, Palomero-Gallagher N. Cyto-, Myelo-, and Receptor Architectonics of the Human Parietal Cortex. NeuroImage. 2001 Jul 1;14(1):S8–20.

49. Eickhoff SB, Rottschy C, Kujovic M, Palomero-Gallagher N, Zilles K. Organizational Principles of Human Visual Cortex Revealed by Receptor Mapping. Cereb Cortex N Y NY. 2008 Nov;18(11):2637–45.

50. Zilles K, Palomero-Gallagher N, Schleicher A. Transmitter receptors and functional anatomy of the cerebral cortex. J Anat. 2004;205(6):417–32.

51. Zilles K, Palomero-Gallagher N, Grefkes C, Scheperjans F, Boy C, Amunts K, et al. Architectonics of the human cerebral cortex and transmitter receptor fingerprints: reconciling functional neuroanatomy and neurochemistry. Eur Neuropsychopharmacol. 2002 Dec 1;12(6):587–99.

52. Morosan P, Schleicher A, Amunts K, Zilles K. Multimodal architectonic mapping of human superior temporal gyrus. Anat Embryol (Berl). 2005 Dec 1;210(5):401–6.

53. Dehaene S, Hauser MD, Duhamel JR, Rizzolatti G. From monkey brain to human brain: A Fyssen foundation symposium. MIT press; 2005.

54. Zilles K, Bacha-Trams M, Palomero-Gallagher N, Amunts K, Friederici AD. Common molecular basis of the sentence comprehension network revealed by neurotransmitter receptor fingerprints. Cortex. 2015 Feb 1;63:79–89.

55. Hansen JY, Shafiei G, Markello RD, Smart K, Cox SML, Nørgaard M, et al. Mapping neurotransmitter systems to the structural and functional organization of the human neocortex. Nat Neurosci. 2022 Nov;25(11):1569–81.

56. Dukart J, Holiga S, Rullmann M, Lanzenberger R, Hawkins PCT, Mehta MA, et al. JuSpace: A tool for spatial correlation analyses of magnetic resonance imaging data with nuclear imaging derived neurotransmitter maps. Hum Brain Mapp. 2021 Feb;42(3):555–66.

57. Nautiyal KM, Hen R. Serotonin receptors in depression: from A to B. F1000Research. 2017 Feb 9;6:123.

58. Seeman P. Schizophrenia and dopamine receptors. Eur Neuropsychopharmacol J Eur Coll Neuropsychopharmacol. 2013 Sep;23(9):999–1009.

59. Quah SKL, McIver L, Roberts AC, Santangelo AM. Trait Anxiety Mediated by Amygdala Serotonin Transporter in the Common Marmoset. J Neurosci Off J Soc Neurosci. 2020 Jun 10;40(24):4739–49.

60. Lydiard RB. The role of GABA in anxiety disorders. J Clin Psychiatry. 2003;64 Suppl 3:21–7.

61. Cipriani A, Furukawa TA, Salanti G, Chaimani A, Atkinson LZ, Ogawa Y, et al. Comparative efficacy and acceptability of 21 antidepressant drugs for the acute treatment of adults with major depressive disorder: a systematic review and network meta-analysis. The Lancet. 2018 Apr;391(10128):1357–66.

62. Huhn M, Nikolakopoulou A, Schneider-Thoma J, Krause M, Samara M, Peter N, et al. Comparative efficacy and tolerability of 32 oral antipsychotics for the acute treatment of adults with multi-episode schizophrenia: a systematic review and network meta-analysis. The Lancet. 2019 Sep 14;394(10202):939–51.

63. Soomro GM, Altman DG, Rajagopal S, Browne MO. Selective serotonin re‐uptake inhibitors (SSRIs) versus placebo for obsessive compulsive disorder (OCD). Cochrane Database Syst Rev [Internet]. 2008 [cited 2022 May 2];(1). Available from: https://www.cochranelibrary.com/cdsr/doi/10.1002/14651858.CD001765.pub3/abstract

64. Geddes JR, Miklowitz DJ. Treatment of bipolar disorder. The Lancet. 2013 May 11;381(9878):1672–82.

65. Moncrieff J, Cooper RE, Stockmann T, Amendola S, Hengartner MP, Horowitz MA. The serotonin theory of depression: a systematic umbrella review of the evidence. Mol Psychiatry. 2022 Jul 20;1–14.

66. Kaltenboeck A, Harmer C. The neuroscience of depressive disorders: A brief review of the past and some considerations about the future. Brain Neurosci Adv. 2018 Jan 1;2:2398212818799269.

67. Kesby JP, Eyles DW, McGrath JJ, Scott JG. Dopamine, psychosis and schizophrenia: the widening gap between basic and clinical neuroscience. Transl Psychiatry. 2018 Jan 31;8(1):1–12.

68. Dean J, Keshavan M. The neurobiology of depression: An integrated view. Asian J Psychiatry. 2017 Jun 1;27:101–11.

69. Harrison PJ, Geddes JR, Tunbridge EM. The Emerging Neurobiology of Bipolar Disorder. Trends Neurosci. 2018 Jan;41(1):18–30.

70. Luvsannyam E, Jain MS, Pormento MKL, Siddiqui H, Balagtas ARA, Emuze BO, et al. Neurobiology of Schizophrenia: A Comprehensive Review. Cureus [Internet]. 2022 Apr 8 [cited 2022 Aug 2];14(4). Available from: https://www.cureus.com/articles/92077-neurobiology-of-schizophrenia-a-comprehensive-review

71. Pauls DL, Abramovitch A, Rauch SL, Geller DA. Obsessive–compulsive disorder: an integrative genetic and neurobiological perspective. Nat Rev Neurosci. 2014 Jun;15(6):410–24.

72. Bell PT, Shine JM. Subcortical contributions to large-scale network communication. Neurosci Biobehav Rev. 2016 Dec 1;71:313–22.

73. Shine JM. Neuromodulatory Influences on Integration and Segregation in the Brain. Trends Cogn Sci. 2019 Jul 1;23(7):572–83.

74. Forstmann BU, de Hollander G, van Maanen L, Alkemade A, Keuken MC. Towards a mechanistic understanding of the human subcortex. Nat Rev Neurosci. 2017 Jan;18(1):57–65.

75. Yeh CH, Jones DK, Liang X, Descoteaux M, Connelly A. Mapping Structural Connectivity Using Diffusion MRI: Challenges and Opportunities. J Magn Reson Imaging. 2021 Jun;53(6):1666–82.

76. Paquola C, Vos De Wael R, Wagstyl K, Bethlehem RAI, Hong SJ, Seidlitz J, et al. Microstructural and functional gradients are increasingly dissociated in transmodal cortices. PLOS Biol. 2019 May;17(5):e3000284.

77. Logothetis NK. What we can do and what we cannot do with fMRI [Internet]. Vol. 453, Nature. Nature Publishing Group; 2008. p. 869–78. Available from: https://www.nature.com/articles/nature06976

78. Yarkoni T, Poldrack RA, Nichols TE, Van Essen DC, Wager TD. Large-scale automated synthesis of human functional neuroimaging data. Nat Methods. 2011 Jun 26;8(8):665–70.

79. Thompson PM, Stein JL, Medland SE, Hibar DP, Vasquez AA, Renteria ME, et al. The ENIGMA Consortium: large-scale collaborative analyses of neuroimaging and genetic data. Brain Imaging Behav. 2014 Jun 1;8(2):153–82.

80. Margulies DS, Ghosh SS, Goulas A, Falkiewicz M, Huntenburg JM, Langs G, et al. Situating the default-mode network along a principal gradient of macroscale cortical organization. Proc Natl Acad Sci U S A. 2016 Nov;113(44):12574–9.

81. Turnbull A, Karapanagiotidis T, Wang HT, Bernhardt BC, Leech R, Margulies D, et al. Reductions in task positive neural systems occur with the passage of time and are associated with changes in ongoing thought. Sci Rep. 2020 Jun 18;10(1):9912.

82. Smallwood J, Bernhardt BC, Leech R, Bzdok D, Jefferies E, Margulies DS. The default mode network in cognition: a topographical perspective. Nat Rev Neurosci. 2021 Aug;22(8):503–13.

83. Murphy C, Wang HT, Konu D, Lowndes R, Margulies DS, Jefferies E, et al. Modes of operation: A topographic neural gradient supporting stimulus dependent and independent cognition. NeuroImage. 2019 Feb 1;186:487–96.

84. Hettwer MD, Larivière S, Park BY, van den Heuvel OA, Schmaal L, Andreassen OA, et al. Coordinated cortical thickness alterations across six neurodevelopmental and psychiatric disorders. Nat Commun. 2022 Nov 11;13(1):6851.

85. Paquola C, Seidlitz J, Benkarim O, Royer J, Klimes P, Bethlehem RAI, et al. A multi-scale cortical wiring space links cellular architecture and functional dynamics in the human brain. PLOS Biol. 2020 Nov 30;18(11):e3000979.

86. Guell X, Schmahmann JD, Gabrieli JD, Ghosh SS. Functional gradients of the cerebellum. Bostan A, Ivry RB, editors. *eLife*. 2018 Aug 14;7:e36652.

87. Tor D. W. NeuroSynth: a new platform for large-scale automated synthesis of human functional neuroimaging data. Front Neuroinformatics. 2011;5.

88. Poldrack RA, Mumford JA, Schonberg T, Kalar D, Barman B, Yarkoni T. Discovering Relations Between Mind, Brain, and Mental Disorders Using Topic Mapping. PLOS Comput Biol. 2012 Oct 11;8(10):e1002707.

89. Thomas Yeo BT, Krienen FM, Sepulcre J, Sabuncu MR, Lashkari D, Hollinshead M, et al. The organization of the human cerebral cortex estimated by intrinsic functional connectivity. J Neurophysiol. 2011 Sep;106(3):1125–65.

90. Spreng RN, Mar RA, Kim ASN. The common neural basis of autobiographical memory, prospection, navigation, theory of mind, and the default mode: a quantitative meta-analysis. J Cogn Neurosci. 2009 Mar;21(3):489–510.

91. Smallwood J, Brown K, Baird B, Schooler JW. Cooperation between the default mode network and the frontal–parietal network in the production of an internal train of thought. Brain Res. 2012 Jan 5;1428:60–70.

92. Langner R, Leiberg S, Hoffstaedter F, Eickhoff SB. Towards a human self-regulation system: Common and distinct neural signatures of emotional and behavioural control. Neurosci Biobehav Rev. 2018 Jul;90:400–10.

93. Preti MG, Van De Ville D. Decoupling of brain function from structure reveals regional behavioral specialization in humans. Nat Commun. 2019 Oct 18;10(1):4747.

94. Liu ZQ, Vázquez-Rodríguez B, Spreng RN, Bernhardt BC, Betzel RF, Misic B. Time-resolved structure-function coupling in brain networks. Commun Biol. 2022 Jun 2;5(1):1–10.

95. Valk SL, Xu T, Paquola C, Park B yong, Bethlehem RAI, Vos de Wael R, et al. Genetic and phylogenetic uncoupling of structure and function in human transmodal cortex. Nat Commun. 2022 May 9;13(1):2341.

96. Amunts K, Lenzen M, Friederici AD, Schleicher A, Morosan P, Palomero-Gallagher N, et al. Broca’s Region: Novel Organizational Principles and Multiple Receptor Mapping. PLOS Biol. 2010 Sep 21;8(9):e1000489.

97. Hettwer MD, Larivière S, Park BY, Heuvel O van den, Schmaal L, Andreassen OA, et al. Coordinated Cortical Thickness Alterations across Psychiatric Conditions: A Transdiagnostic ENIGMA Study [Internet]. medRxiv; 2022 [cited 2022 Apr 1]. p. 2022.02.03.22270326. Available from: https://www.medrxiv.org/content/10.1101/2022.02.03.22270326v1

98. Mueller S, Wang D, Fox MD, Yeo BTT, Sepulcre J, Sabuncu MR, et al. Individual variability in functional connectivity architecture of the human brain. Neuron. 2013 Feb 6;77(3):586–95.

99. Vos de Wael R, Benkarim O, Paquola C, Lariviere S, Royer J, Tavakol S, et al. BrainSpace: a toolbox for the analysis of macroscale gradients in neuroimaging and connectomics datasets. Commun Biol. 2020 Dec;3(1):1–10.

100. Royer J, Rodríguez-Cruces R, Tavakol S, Larivière S, Herholz P, Li Q, et al. An Open MRI Dataset For Multiscale Neuroscience. Sci Data. 2022 Sep 15;9(1):569.

101. Neve KA, Seamans JK, Trantham-Davidson H. Dopamine Receptor Signaling. J Recept Signal Transduct. 2004 Jan 1;24(3):165–205.

102. Zarkali A, McColgan P, Leyland LA, Lees AJ, Rees G, Weil RS. Organisational and neuromodulatory underpinnings of structural-functional connectivity decoupling in patients with Parkinson’s disease. Commun Biol. 2021 Jan 19;4(1):1–13.

103. Larivière S, Vos de Wael R, Hong SJ, Paquola C, Tavakol S, Lowe AJ, et al. Multiscale Structure–Function Gradients in the Neonatal Connectome. Cereb Cortex. 2020 Jan 10;30(1):47–58.

104. Tian Y, Margulies DS, Breakspear M, Zalesky A. Topographic organization of the human subcortex unveiled with functional connectivity gradients. Nat Neurosci. 2020 Nov;23(11):1421–32.